# EUCLID-OMNI: A UNIFIED NEURO-SYMBOLIC FRAMEWORK FOR GEOMETRY PROBLEM SOLVING

## ABSTRACT

Euclidean geometry presents a compelling testbed for AI reasoning capabilities, requiring seamless integration of diagram understanding, logical deduction, and algebraic computation. Existing systems have either been narrowly scoped or struggled with challenging problems. We introduce Euclid-Omni, a unified neuro-symbolic framework that combines a formal geometry system with Large (Vision)–Language Models (LLMs and VLMs) to address both calculation- and proving-style problems across formal and natural languages, up to Olympiad-level difficulty. At its core, we develop Euclidea, a versatile geometry symbolic solver that automatically generates human-readable reasoning steps through logical deduction and algebraic solving. On top of this, we implement a comprehensive data generation pipeline that synthesizes symbolic problems, renders diagrams, and translates problems into natural language, yielding large-scale, diverse datasets for training LLMs and VLMs in different reasoning settings. Experiments on multiple benchmarks demonstrate that Euclidea can tackle a broader range of problems than prior symbolic systems. Our trained VLMs achieve superior results on calculation tasks, while combining LLMs with Euclidea remains competitive with state-of-the-art systems on Olympiad-level theorem proving problems, despite using orders of magnitude less compute and data.

## 1 INTRODUCTION

Plane geometry, dating back to Euclid's *Elements*, has long been a cornerstone of mathematics education around the world. More recently, it has become a key testbed for AI and large (vision)-language models (LLMs and VLMs) in mathematical reasoning (Zhao et al., 2025b; Ma et al., 2025), particularly for AI to compete in the International Mathematical Olympiad (IMO). State-of-the-art systems such as AlphaGeometry (Trinh et al., 2024; Chervonyi et al., 2025) and Seed-Geometry (Chen et al., 2025) have matched the performance of IMO gold medalists, following a common framework: (i) a symbolic engine that exhaustively derives new propositions, (ii) a language model that proposes auxiliary constructions to be combined with the symbolic engine, and (iii) a data generator that synthesizes large-scale theorems and proofs to train the language model.

Despite their impressive performance in the IMO, these systems face significant limitations in both scope and accessibility. First, they are narrowly tailored to competition-style theorem proving and lack support for complex algebraic computations, thereby neglecting calculation-based problems that are equally common in plane geometry. Moreover, they cannot always produce human-like solution steps or incorporate modalities such as natural language or visual diagrams, limiting their relevance to applications like mathematics education. By contrast, more general-purpose systems cover only a limited set of geometric theorems and remain far below IMO-level performance (Chen et al., 2021b; Lu et al., 2021; Chen et al., 2022). Second, existing IMO-level systems impose substantial barriers to accessibility. Their training pipelines demand enormous computational resources—for example, AlphaGeometry (Trinh et al., 2024) generated 500M training examples using 100K CPUs—and none has released its data-generation pipeline or datasets. In contrast, lots of publicly available datasets remain small, limited in diversity, and poorly stratified by difficulty, which makes them inadequate for training modern LLMs and VLMs (Chen et al., 2021b; Lu et al., 2021; Cao & Xiao, 2022; Chen et al., 2022). As a result, while current systems mark important progress, they remain narrow in scope and difficult to extend as foundations for broader research.

To address these issues, we introduce Euclid-Omni, a unified neuro-symbolic framework that integrates LLMs and VLMs with a versatile symbolic system for diverse geometric reasoning tasks. Central to Euclid-Omni is Euclidea, a Python-based symbolic system that unifies the representation and reasoning of plane geometry problems for both proving and calculation tasks. Euclidea encodes geometric conditions from text and spatial information from diagrams, and performs reasoning by integrating a deductive database of human-like inference rules with an advanced algebraic engine for equation solving. This enables it to automatically solve a problem, up to the level of IMO problems. Building on Euclidea, we design a comprehensive pipeline generating training data for LLMs and VLMs: a synthetic problem generator produces problems from scratch, a diagram renderer draws corresponding visual diagrams, and Euclidea outputs symbolic solutions. These symbolic problems and solutions are then translated into natural language through a hybrid strategy, where rule-based templates first map the symbolic semantics into aligned natural-language form, after which off-the-shelf LLMs refine the text into fluent language. The pipeline is highly configurable, enabling the creation of datasets tailored to specific goals, such as final numerical answers, stepwise proofs, or auxiliary constructions, across difficulty levels from elementary to IMO.

We conduct extensive experiments to evaluate Euclid-Omni across diverse task setups. To begin, we benchmark the symbolic engine, Euclidea, on three widely used datasets: Geometry3K (Lu et al., 2021), JGEX-AG-231 (Trinh et al., 2024), and IMO-AG-30 (Trinh et al., 2024), which together span both calculation and proving tasks. Euclidea outperforms existing formal geometry systems by solving more problems and generating solutions that are not only more accurate but also more human-readable. Notably, it is the first to directly solve two additional IMO problems that all previous symbolic solvers failed to handle.

In addition, we train a VLM on our synthetic calculation-oriented dataset and evaluate them on benchmark problems from GeoQA (Chen et al., 2021b), Geometry3K (Lu et al., 2021), Math-Vista (Lu et al., 2024), and MathVerse (Zhang et al., 2024c). The model operates directly on natural language and diagram inputs, without relying on symbolic representations or the symbolic engine. Despite using less training data than existing approaches, our VLMs achieve comparable or superior performance across these benchmarks.

Finally, we train an LLM on our synthetic auxiliary-construction datasets and integrate it with Euclidea to tackle challenging Olympiad-level geometry problems from JGEX-AG-231 (Trinh et al., 2024) and IMO-AG-30 (Trinh et al., 2024). Our model consistently outperforms existing API-based baselines and achieves performance comparable to state-of-the-art systems (Trinh et al., 2024) trained on reduced data budgets—while using orders of magnitude less training data.

In summary, we introduce Euclid-Omni, a unified framework for solving geometry problems. It is the first system to support diverse modes of geometric reasoning—including both proof and calculation tasks, as well as formal and natural language inputs—while achieving performance at the IMO level. We will release the complete framework to facilitate future research and practical applications.

## 2 RELATED WORK

In this section, we review three areas most relevant to our work: formal plane geometry systems, geometry datasets and benchmarks, and learning-based approaches for geometric reasoning.

**Symbolic Approaches.** Classical formal geometry solvers follow two main paradigms (Chou & Gao, 2001): *synthetic deduction* and *algebraic computation*. Synthetic methods, such as the deductive database approach (Chou et al., 2000; Ye et al., 2011), employ forward chaining to systematically apply geometric rules and derive new facts, but they falter on problems requiring complex algebraic manipulation. Algebraic methods, including Gröbner basis (Kutzler & Stifter, 1986) and Wu's method (Wu, 1986), encode geometric relations as polynomial equations and solve them algebraically, offering strong reasoning power but often producing proofs that are difficult to interpret. Hybrid systems, such as NGS (Chen et al., 2021b) and Inter-GPS (Lu et al., 2021) for calculation, LeanEuclid (Murphy et al., 2024) and DD+AR (Trinh et al., 2024) for theorem proving, attempt to combine deductive and algebraic reasoning, yet remain specialized in certain task types with limited generality. Other frameworks, including FormalGeo (Zhang et al., 2023b) and PyEuclid (Li et al., 2025), pursue a unified approach, but still struggle to scale efficiently to Olympiad-level problems.

**Datasets and Benchmarks.** Many geometry datasets are derived from textbooks, exercises, and competitions, where problems are paired with manually constructed symbolic formulations (Cao & Xiao, 2022; Chen et al., 2022; Zhang et al., 2023a;b). Examples include calculation-oriented datasets such as GeoQA (Chen et al., 2021b) and Geometry3K (Lu et al., 2021), as well as theorem proving datasets such as UniGeo (Chen et al., 2022), JGEX-AG-231 (Trinh et al., 2024), and IMO-AG-30 (Trinh et al., 2024). Due to the high cost of manual annotation, these resources remain relatively small in scale. To expand coverage, recent work (Gao et al., 2023; Zhang et al., 2024d) has leveraged LLMs and VLMs to generate larger datasets by augmenting existing problems, though diversity remains bounded by the underlying sources. In parallel, several synthetic pipelines (Kazemi et al., 2023; Deng et al., 2024; Pan et al., 2025; Fu et al., 2025; Wu et al., 2025) generate symbolic problems using basic geometric primitives and predicates, but the resulting problems are generally constrained in both difficulty and variety. Other benchmarks (Zhang et al., 2024b; Wang et al., 2024; Qiao et al., 2024; Xu et al., 2025), including MathVista (Lu et al., 2024) and MathVerse (Zhang et al., 2024c), collect a broad variety of geometry problems in natural language to evaluate the reasoning abilities of VLMs. Besides solving geometry problems directly, auxiliary datasets have also been introduced for related tasks such as autoformalization (Murphy et al., 2024), diagram parsing (Hao et al., 2022), diagram understanding (Huang et al., 2025), and geometric image generation (Cai et al., 2024).

**Learning-Based Methods.** Recent advances in LLMs and VLMs have spurred a wave of learning-based approaches to geometric reasoning (Zhao et al., 2025b; Ma et al., 2025). One line of work adopts *neuro-symbolic* methods that operate over symbolic representations (Chen et al., 2021b; Lu et al., 2021; Chen et al., 2022; Peng et al., 2023; Gao et al., 2023; Wu et al., 2024; Trinh et al., 2024; Zhang et al., 2024a; Duan et al., 2024; Zhao et al., 2025a; Zhang et al., 2025; Ping et al., 2025): these approaches leverage LLMs or VLMs to generate solution steps in symbolic form and delegate execution to a solver, ensuring both correctness and interpretability. For example, Inter-GPS (Lu et al., 2021) predicts program sequences to compute numerical quantities, while AlphaGeometry (Trinh et al., 2024) predicts auxiliary constructions and integrates them with its inference engine DD+AR for theorem proving. Such systems enable more faithful reasoning but rely heavily on solver design and reasoning capabilities. By contrast, *purely neural* methods attempt to reason directly in natural language (Gao et al., 2023; Xu et al., 2024; Deng et al., 2024; Wu et al., 2025), typically targeting calculation-style problems with easily verifiable answers and using chain-of-thought reasoning to produce step-by-step solutions. While effective on simpler tasks, these approaches often generate hallucinated reasoning steps and are difficult to verify, limiting their reliability for theorem proving and more complex scenarios. Some methods (Ning et al., 2023; Li et al., 2023; 2024; Xia et al., 2024; Cho et al., 2025) also aim to enhance reasoning by aligning VLMs with stronger visual diagram understanding. Nevertheless, no unified framework currently exists for training LLMs and VLMs that can flexibly support diverse geometry tasks across both formal and natural languages.

# 3 EUCLID-OMNI

This section first introduces the proposed formal geometry solver Euclidea, and then presents the Euclid-Omni pipeline for training LLMs and VLMs across both calculation and proving tasks.

## 3.1 EUCLIDEA: SYMBOLIC SOLVER FOR PROVING AND CALCULATION PROBLEMS

Euclidea is a Python-based formal plane geometry system that encodes information from both text and diagrams. Its reasoning engine integrates deductive inference with algebraic computation, enabling it to automatically produce human-readable solutions for both numerical calculation and theorem proving, up to the level of IMO problems. An overview of Euclidea is shown in Figure 1.

**Problem Formalization.** Euclidea formalizes plane geometry by unifying two established approaches to geometric representation (Chou et al., 2000; Avigad et al., 2009). Specifically, it treats points (e.g., `a`, `b`) as basic primitives, while other geometric objects (e.g., lines, triangles) are defined in terms of points (Chou et al., 2000). Each problem, including its goal and accompanying diagram, is formalized as a collection of relations, which fall into two categories (Avigad et al., 2009):

- *Metric relations* encode quantitative properties, such as `Perpendicular(a,b,c,d)` (line `ab` is perpendicular to line `cd`) or `Angle(a,b,c)=`$\pi/2$ ($\angle abc = \pi/2$). These can be expressed as algebraic equations of geometric quantities such as lengths, angles, ratios, and areas.

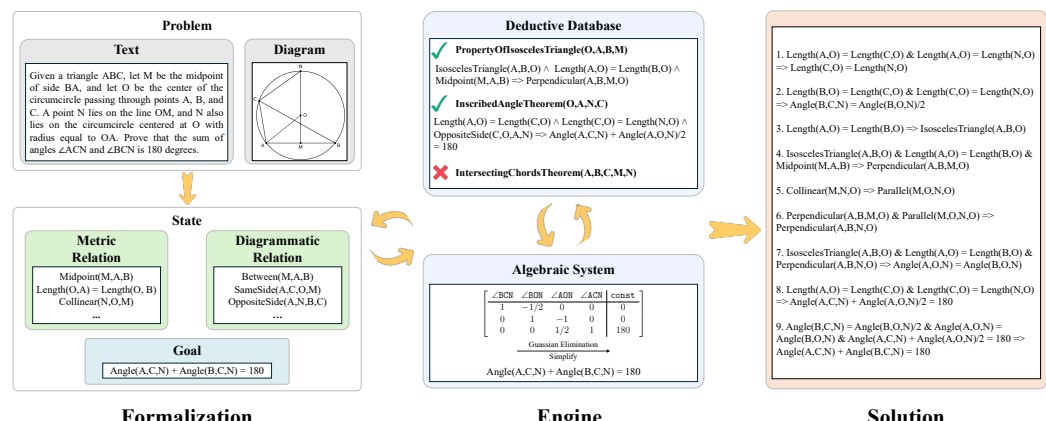

Figure 1: A working example of Euclidea on a proving problem.

- *Diagrammatic relations* capture topological configurations observable directly from diagrams, such as SameSide(a,b,c,d) (points a and b lie on the same side of line cd).

Metric relations are typically explicit in the problem statement or derived via geometric theorems, while diagrammatic relations are implicit but can be extracted from the diagram. We define the problem *state* as the set of current derived metrics and diagrammatic relations together with the goal. Unlike *full-angle* formalization (Chou et al., 2000), which represents angles using pairs of lines, Euclidea encodes them using three points, making the representation closer to human. Moreover, it incorporates diverse diagrammatic relations, ensuring that the topological structure is faithfully captured. This design also eliminates the need to explicitly enumerate additional objects such as lines and circles during formalization (Avigad et al., 2009). Further details are provided in Appendix A.1.

**Reasoning Engine.** Given the current state, Euclidea combines a deductive database with an algebraic system to calculate or prove the target goal. The deductive component extends existing approaches (Chou et al., 2000; Ye et al., 2011; Trinh et al., 2024) with a richer and more fine-grained set of inference rules defined over our formal representations, which are systematically enumerated to identify applicable theorems. For example, the Angle Bisector Theorem can be formalized as:

```
AngleBisectorTheorem(a,b,c,d):  Angle(d,a,b) = Angle(d,a,c) ∧
Collinear(d,b,c) ∧ Between(d,b,c) ∧ Not(Collinear(a,b,c)) ⇒
Length(d,b)/Length(d,c) = Length(a,b)/Length(a,c)
```

where point d lies on line bc and on the angle bisector of ∠*bac*. To mimic human-like reasoning, inference rules are categorized as *intuitive*, *basic*, or *complex* according to predefined difficulty. Enumeration follows this order, ensures that straightforward relations are derived with simpler rules before resorting to more sophisticated reasoning. To avoid redundant permutations of equivalent rules (e.g., AngleBisectorTheorem(a,b,c,d) vs. AngleBisectorTheorem(a,c,b,d)), we impose a lexical partial order over the variable assignment. An SQL database (Gaffney et al., 2022) is used to efficiently encode and enumerates applicable rules, after which all newly derived relations and equations are added to the state. This design enables seamless extension of the system with new inference rules, without the need to manually implement or optimize dedicated enumerators for each theorem. Additional implementation details can be found in Appendix A.3.

Complementing the deductive database, Euclidea integrates a symbolic algebraic system built on SymPy (Meurer et al., 2017) to simplify equations and solve for unknown quantities. Inspired by DD+AR (Trinh et al., 2024), equations are categorized into four types: (i) *angle-based* (fixed angle, angle sum, and angle ratio), (ii) *length-based* (fixed length, length sum, and length ratio), (iii) *length-ratio-based* (fixed length, length ratio, and equalities between ratios or between an area and the product of two lengths), and (iv) *complex* (e.g., trigonometric or higher-order polynomial relations). The first two types can be transformed into a linear system $A\mathbf{x} = \mathbf{b}$, where $\mathbf{x}$ is a vector of geometric quantities and $A$ and $\mathbf{b}$ denote the corresponding coefficients and constants, which is then solved via Gaussian elimination. The third type can be reduced to a log-linear form and solved in the same way after transformation. Simplified results from these three types are merged into a unified system,

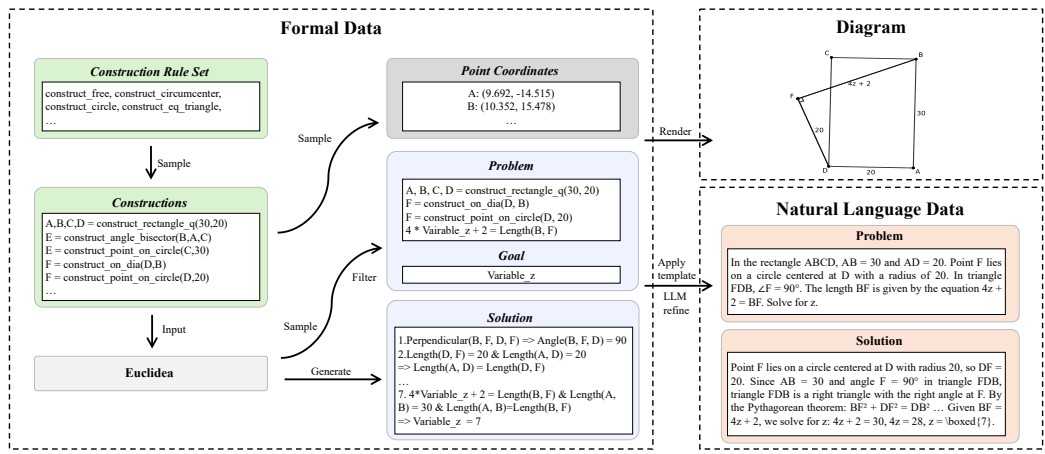

Figure 2: A working example of Euclid-Omni generating data for a calculation problem.

and all newly derived equations of these types are added back into the state. For complex equations, Euclidea leverages the results of the first three types for simplification and substitution, reducing many cases to single- or double-variable equations that can be further simplified into new relations and directly solved for unknown quantities. Some examples are provided in the Appendix A.4. Euclidea iteratively invokes the deductive database and the algebraic system in tandem, with each component reinforcing the other, thereby incrementally expanding the state with new relations. A problem is solved once the goal is either derived in the relations or computed to a numerical value.

**Solution Generation.** To generate human-readable reasoning traces, each relation produced by the deductive database is labeled with its originating inference rule and associated conditions. For each equation $e$ solved via Gaussian elimination, we cast the tracking process as an optimization problem:

$$\min_{\mathbf{z}} \ \|\mathbf{z}\|_t, \quad \text{s.t.} \quad [A \mid \mathbf{b}]^\top \mathbf{z} = \mathbf{c},$$

where $[A \mid \mathbf{b}]$ is the augmented coefficient matrix of the linear system $A\mathbf{x} = \mathbf{b}$, $\mathbf{c}$ is the coefficient vector of the query equation $e$, $\mathbf{z}$ denotes the coefficients for equations contributing to the query, and $t$ specifies the chosen norm (0 or 1) to promote sparsity and ensure minimal traced equations. Numerical optimization is performed using PySCIPOpt (Maher et al., 2016). For quantities derived from complex equations, Euclidea records the original complex equation together with the substituted equations from the previous linear systems. Starting from the goal, Euclidea recursively traces its dependencies until all are grounded in the initial relations. This process yields a dependency graph whose root represents the goal and whose leaf nodes correspond to the original conditions. A post-order traversal of this graph then produces a structured sequence of reasoning steps, formatted into a human-readable solution to enhance interpretability.

## 3.2 Synthetic Data Generation and Model Training

Based on the geometric formalization and reasoning of Euclidea, Euclid-Omni provides a unified framework that integrates a synthetic problem generator, diagram renderer, and natural language translator to produce large-scale, diverse training data with flexible configurations for different geometry tasks. An overview of Euclid-Omni is shown in Figure 2.

**Synthetic Problem Generation.** To create diverse instances, Euclid-Omni synthesizes geometry problems *from scratch* with full control over their structure. Inspired by AlphaGeometry (Trinh et al., 2024), we extend and enrich a library of artificial *construction rules*, each corresponding to a ruler-and-compass operation that bundles a set of condition and conclusion relations, thereby enabling efficient problem generation. For example, the rule `x = construct_foot(a,b,c)`, which constructs the foot of the perpendicular from point a to line bc, is formalized as:

`∃ x, Not(Collinear(a,b,c)) ⇒ Perpendicular(x,a,b,c) ∧ Collinear(x,b,c)`

We categorize construction rules into four types: *independent*, *deterministic*, *non-deterministic*, and *diagrammatic*. An independent rule has no conditions (e.g., constructing an isolated square). A

deterministic rule uniquely determines point coordinates (e.g., constructing the foot of a perpendicular). A non-deterministic rule admits multiple valid placements (e.g., constructing a point on an angle bisector). Diagrammatic rules impose additional topological constraints (e.g., requiring two points to lie on the same side of a line). In problem generation, we begin with independent rules and then sample either deterministic or non-deterministic rules, optionally augmented with diagrammatic constraints. For non-deterministic rules, we allow at most two such constructions if the resulting points can be resolved by their intersection. For each applied construction rule, we first sample exact numerical point coordinates that satisfy its conditions, and add the corresponding conclusions to Euclidea. To support calculation-style problems, we further parameterize a subset of rules with explicit quantities such as lengths or angles. For example, `a,b,c,d =` `construct_square_q(l)` constructs a square `abcd` with side length `l`, represented as:

$$\exists \text{ a,b,c,d, True} \Rightarrow \text{Square(a,b,c,d)} \wedge \text{Length(a,b)} = l$$

It is worth noting that even when applying the same sequence of construction rules, the randomly sampled initial points can yield different coordinates and diagrammatic relations across runs. An illustrative example is provided in Appendix A.2.

Given a sampled problem in the form of construction rules, we derive additional diagrammatic relations from the sampled point coordinates and combine them with the conclusions of each rule as input to Euclidea. Euclidea then infers all possible new relations and produces the corresponding solution steps. From this expanded set of relations, we can filter and select specific ones as target goals according to the requirements of different tasks. For each selected goal, we also trace back to its minimal set of construction rules from the associated solution, ensuring that no redundant constructions remain and avoiding unnecessary complexity in the problem.

**Diagram Rendering.** We implement a diagram visualizer that renders geometric objects such as segments and circles to illustrate each sampled problem given its numerical point coordinates. For every construction rule, we specify the required objects and quantities, which are then drawn on a canvas using Matplotlib (Hunter, 2007). For example, the rule `x = construct_foot(a,b,c)` produces segments `bc`, `xa`, `xb`, and `xc`, along with the right angle $\angle axb$.

**Natural Language Translation.** To train models to perform geometric reasoning in natural language, Euclid-Omni translates symbolic problems and solutions into fluent text. Our preliminary studies show that directly prompting off-the-shelf LLMs for this task requires specifying the entire geometric formalization and its semantics, leading to lengthy prompts and no guarantee of correctness or validation. To address this, we adopt a hybrid strategy (Huang et al., 2025). We first build a library of manually verified natural language templates for each construction rule in the problem and relation in the solution step. Given a symbolic problem and solution, we parse them and instantiate an appropriate template to obtain an aligned natural language version. An LLM is then employed to refine these outputs into diverse and fluent problem statements and solutions. Examples of templates and prompts are provided in the Appendix B.1.

**Task Configuration.** With Euclid-Omni, we can generate both formal and natural language problems, solutions, and corresponding diagrams. The pipeline allows users to flexibly configure each component to produce data tailored to different tasks. In this paper, we focus on two representative settings for geometry problem solving: (i) solving calculation-style problems in natural language with VLMs, (ii) predicting auxiliary constructions in formal language with LLMs, which can then be combined with symbolic solvers to address Olympiad-level problems.

For the first task, the problem generator samples construction rules with or without quantitative parameterization, filtering target goals to those involving lengths, angles, or areas. Variable-based formulations are also supported by defining linear equations over lengths or angles, such as $x + 10° = \angle abc$, and treating the variable as the goal. Each problem is synthesized using 3–5 construction rules sampled uniformly. Angles are drawn from the ranges $[10°, 80°]$ and $[100°, 170°]$, with special emphasis on notable values such as $15°, 30°, 45°, 60°, 90°,$ and $120°$. Lengths are sampled from the interval $[1, 15]$, with an additional random scaling factor from $[1, 10]$. To support multiple-choice formats in existing benchmarks, we adapt LLM prompts during natural language translation to generate plausible distractors of comparable scale to the correct answer. For training, we retain the generated natural language problem, its corresponding diagram, the step-by-step solution, and its final answer/choice. The supervised fine-tuning

Table 1: Number of solved problems by different formal geometry systems. – indicates that a solver does not support the given task or formalization. $^{\dagger}$ indicates results taken directly from prior works.

| Task | Dataset | #Problems | Formal Geometry System | | | | |
|---|---|---|---|---|---|---|---|
| | | | Inter-GPS$^{\dagger}$ | PyEuclid$^{\dagger}$ | DD+AR$^{\dagger}$ | Newclid | Euclidea |
| Calculation | Geometry3K | 601 | 426 | 567 | – | – | **595** |
| Proving | JGEX-AG-231 | 231 | – | 202 | 198 | 188 | **207** |
| | IMO-AG-30 | 30 | – | – | 14 | 14 | **16** |

template is: `Inputs: <diagram> <natural language problem> Outputs: <natural language solution> \boxed{<final answer/choice>}`

For the second task, the problem generator invokes Euclidea after each applied construction rule to check whether it can derive relations *not involving* the newly constructed points. If such a relation is derived, the corresponding rule is marked as an auxiliary construction necessary for establishing that relation. For example, suppose we add the construction `g = construct_foot(d,e,f)` and then derive the relation `be = ef`. Since this relation involves only the points `b,e,f` and does not depend on the construction of `g`, this step is identified as an auxiliary construction for the goal `be = ef`. The derived relation is then set as the goal, and Euclidea is rerun to generate a more concise proof and identify a minimal set of both necessary and auxiliary construction rules. For this setting, we sample 8–10 construction rules and filter goals to common Olympiad-style targets such as midpoint, collinearity, similarity, congruence, concyclicity, and equality of lengths or angles. Following prior work (Zhang et al., 2024a), only the formal problem statement and the auxiliary constructions are retained for training. The training template for this task is: `Inputs: <formal problem> Outputs: <formal auxiliary constructions>`

We provide several examples of synthetic instances of these two tasks in Appendix B.2. Note that the Euclid-Omni pipeline can also be configured to generate data for other useful tasks such as autoformalization, diagram parsing, and diagram understanding. We discuss some of these potential applications in Section 5 and leave them as future directions for the community to explore.

## 4 EXPERIMENTS

In this section, we conduct a series of experiments to evaluate Euclidea and Euclid-Omni on both calculation- and proof-oriented geometry tasks in formal and natural language settings.

### 4.1 EVALUATION OF EUCLIDEA AGAINST FORMAL GEOMETRY SYSTEMS

**Setup.** We evaluate Euclidea against open-source formal geometry systems on three datasets: (i) Geometry3K (Lu et al., 2021), a collection of SAT-style problems for calculation tasks; (ii) JGEX-AG-30 (Trinh et al., 2024), which includes well-known theorems and Olympiad-level problems from textbooks; and (iii) IMO-AG-30 (Trinh et al., 2024), which consists of IMO problems from 2000–2022. For Geometry3K, we adopt the formalization introduced in PyEuclid (Li et al., 2025) with minor modifications, and we additionally correct several manually labeled errors in the original logical forms. For the other two datasets, we use the original formalization almost without modification. We compare Euclidea with four open-source symbolic systems: Inter-GPS (Lu et al., 2021) and PyEuclid (Li et al., 2025) for calculation tasks, DD+AR (Trinh et al., 2024), Newclid (Sicca et al., 2024), and PyEuclid for proving tasks. Following the evaluation protocol of PyEuclid, a numerical answer is considered correct if it differs by less than 2% from the labeled solution, since the ground truth may contain rounding inaccuracies. For proving tasks, the system must successfully generate a valid proof. A time limit of 600s is applied to all problems.

**Results.** Table 1 summarizes the performance of different formal geometry solvers across three datasets. Euclidea consistently outperforms all existing systems across all benchmarks. Remarkably, it solves 99% of problems in Geometry3K and successfully tackles two additional challenging IMO problems compared to prior work. The few remaining unsolved problems fall into three categories: (i) problems that cannot be formalized within Euclidea, (ii) problems with more than 15 points that

Table 2: Performance of different VLMs on four benchmarks, reported as the percentage (%) of correctly solved problems. Baseline results are taken directly from prior works when available.

| Model | #Train | GeoQA | Geometry3K | MathVista | MathVerse |
|---|---|---|---|---|---|
| G-LLaVA-7B | 117K | 64.2 | - | 53.4 | - |
| MAVIS-7B | 834K | - | - | 64.1 | 27.9 |
| Qwen2.5-VL-7B | - | 69.4 | 56.4 | 72.2 | 44.1 |
| Qwen2.5-VL-7B + NeSyGeo | 100K | 71.8 | - | - | 46.7 |
| Qwen2.5-VL-7B + GeoGen | 224K | 77.6 | 58.4 | 74.0 | - |
| Qwen2.5-VL-7B + TR-COT | 183K | 79.2 | - | 74.5 | - |
| Ours | 20K | 76.6 | 61.0 | 74.7 | 51.0 |

yield prohibitively large search spaces and cause timeouts, and (iii) problems requiring auxiliary constructions. Beyond raw performance gains, Euclidea also produces higher-quality proofs. As detailed in the Appendix C.1, our generated proofs are more human-readable and better aligned with visual diagrams than the full-angle-based proofs (Chou et al., 1996) produced by DD+AR and Newclid, while remaining significantly more compact than those generated by PyEuclid.

## 4.2 EVALUATION OF EUCLID-OMNI ON NATURAL-LANGUAGE CALCULATION PROBLEMS

**Setup.** We use Euclid-Omni to synthesize 10K problem instances for training. These problems are translated into natural language using Gemini 2.5 Flash (Comanici et al., 2025) as the LLM component of Euclid-Omni. We then train Qwen2.5-VL (Bai et al., 2025) on this dataset for one epoch with 4×H100 GPUs, implemented via LLaMA-Factory (Zheng et al., 2024). Following prior works (Pan et al., 2025; Deng et al., 2024), to better match the out-of-distribution diagrams present in existing benchmarks, we augment our training data by incorporating 10K random samples from the Geo170K dataset (Gao et al., 2023). We then train our model on the combined set of 20K samples, ensuring better alignment with the benchmark diagram distribution.

For evaluation, we compare our trained VLM against existing baselines: G-LLaVA (Gao et al., 2023), MAVIS (Zhang et al., 2024d), Qwen2.5-VL (Bai et al., 2025), with a line of works which employ Qwen2.5-VL as the base model, namely GeoGen (Pan et al., 2025), TR-COT (Deng et al., 2024), and NeSyGeo (Wu et al., 2025). All of them are trained solely via supervised fine-tuning on synthesized datasets. Experiments are conducted on four benchmarks: GeoQA (Chen et al., 2021b), Geometry3K (Lu et al., 2021), the plane-geometry subsets of MathVista (testmini split) (Lu et al., 2024) and MathVerse (vision-intensive split) (Zhang et al., 2024c). Following prior work, we report accuracy as multiple-choice accuracy when answer options are available, and otherwise evaluate by checking whether the predicted result matches the ground-truth numerical value.

**Results.** Table 2 summarizes the performance of various VLMs across the evaluated datasets. Our trained model achieves state-of-the-art results on three out of four benchmarks and surpasses the base model by an average margin of $5.3\%$, while remaining competitive on the GeoQA dataset. Importantly, our approach requires orders of magnitude less training data than existing baselines. In addition, synthetic data, particularly diagrams, differ substantially from those in evaluation benchmarks, highlighting the strong generalization ability enabled by our synthetic data. For additional insights, Appendix C.2 presents qualitative comparisons between solutions produced by our models and those from the base model.

## 4.3 COMPARISON OF THE HYBRID SYSTEM ON FORMAL OLYMPIAD-LEVEL PROBLEMS

**Setup.** We generate 100K training samples involving auxiliary constructions and train Qwen2.5-Math-7B Yang et al. (2024) as the base model. During inference, we employ beam search to propose auxiliary constructions, ranking candidates by log probability and iteratively expanding the search whenever the solver fails to reach the goal. The search is configured with a branching factor of 32, a beam size of 128, and a maximum depth of 4. For baselines, we compare against AlphaGeometry (Trinh et al., 2024) as well as two API-based models, GPT-4o and Gemini 2.5 Flash, selected for their strong reasoning capabilities and cost efficiency under the high query volume induced by beam search. Both GPT-4o and Gemini 2.5 Flash are prompted with the formal semantics of DD+AR (Trinh et al., 2024) and Euclidea to generate auxiliary constructions. Experiments are con-

ducted on two established benchmarks: JGEX-AG-231 (Trinh et al., 2024) and IMO-AG-30 (Trinh et al., 2024), with a timeout of 90 minutes per problem, consistent with the standard time in IMO.

**Results.** Table 3 reports the results on Olympiad-level benchmarks. Existing API-based models provide only marginal gains, typically solving relatively simple problems that require a single auxiliary construction. In contrast, our hybrid system solves 223 problems on JGEX-AG-231 and 22 on IMO-AG-30. Remarkably, while AlphaGeometry is trained on 100M problems, our system achieves competitive performance with only 100K training sam-

Table 3: Number of solved problems across two benchmarks.

| Engine | Method | #Train | JGEX-AG-231 | IMO-AG-30 |
|---|---|---|---|---|
| DD+AR | GPT-4o | - | 213 | 17 |
| | Gemini 2.5 Flash | - | 216 | 17 |
| | AlphaGeometry | 100M | 228 | 25 |
| | AlphaGeometry | 20M | - | 21 |
| Euclidea | GPT-4o | - | 213 | 17 |
| | Gemini 2.5 Flash | - | 213 | 17 |
| | Ours | 100K | 223 | 22 |

ples, and surpasses AlphaGeometry when trained on 20M samples. This highlights both the efficiency and data-effectiveness of our approach. Furthermore, our solver often discovers auxiliary constructions distinct from those of AlphaGeometry and may require fewer steps to reach a solution. Detailed examples are provided in Appendix C.3.

## 5 LIMITATIONS AND FUTURE WORK

While Euclidea and Euclid-Omni represent a promising unified approach to automated geometry problem solving across diverse settings, several avenues remain open for further improvement.

**Toward More Expressive and Human-Like Euclidea.** Although the current design of Euclidea aims to mimic human reasoning and produce human-interpretable solutions, the resulting proofs can be lengthy, particularly for Olympiad-level problems, and differ substantially from those written by IMO contestants. A natural extension is to enrich the deductive database with more sophisticated rules (e.g., well-known theorems) such as Menelaus' or Desargues' theorem. This could yield more compact proofs while broadening the range of solvable problems. Another direction is to enhance the algebraic system by incorporating projective or inversive geometry, thereby increasing expressivity. Finally, since the formalization underlying Euclidea closely resembles LeanEuclid (Murphy et al., 2024), a compelling avenue is to ground its representations in Lean. This would allow Euclidea to function as an automatic tactic and integrate with existing libraries (Mathlib community, 2020; Song et al., 2025), bridging automated geometry solvers with general-purpose proof assistants.

**Toward Better Design and Broader Usage of Euclid-Omni.** Several opportunities remain to refine the design of Euclid-Omni and expand its applications. Currently, construction rules are sampled uniformly, without leveraging human priors or distributions observed in existing problems. Future work could incorporate statistics from existing problems (Zhang et al., 2024a) or exploit structural priors such as diagram symmetries to better align generated problems with downstream tasks. With greater computational resources, scaling Euclid-Omni may also produce larger problem sets closer in complexity to IMO-level benchmarks (Trinh et al., 2024; Zhang et al., 2024a; Chervonyi et al., 2025). In addition, while current Euclid-Omni trains LLMs and VLMs via supervised finetuning on synthetic data, there is significant potential to explore reinforcement learning, using Euclidea to provide verifiable feedback during training. Beyond the tasks studied here, Euclid-Omni has broader potential: it could support autoformalization and problem solving to produce verifiable solutions from natural-language descriptions (useful for pedagogy), serve as a pretraining resource for VLMs to improve vision–language alignment and diagram understanding, or enable training models that learn to generate code for diagram generation directly from natural-language problems.

## 6 CONCLUSION

We introduced Euclid-Omni, a unified neuro-symbolic framework that integrates a novel formal system Euclidea with LLMs and VLMs for geometric reasoning. The pipeline unifies symbolic problem and solution generation, diagram construction, and natural language translation, and can be flexibly configured for a wide range of geometry tasks. Experiments demonstrate that Euclidea and Euclid-Omni achieve strong performance across multiple datasets, underscoring their effectiveness for automated geometry problem solving. Looking forward, we hope this work will serve as a foundation for extending the capabilities of LLMs and VLMs beyond geometry to broader applications.

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

# A    MORE DETAILS OF EUCLIDEA

## A.1    PROBLEM FORMALIZATION

Following System-E (Avigad et al., 2009), all relations in Euclidea fall into two categories: *metric conditions* and *diagrammatic conditions*. A diagrammatic condition asserts certain topological configurations of the diagram, which are encoded as predicate propositions, such as `Between(a,b,c)`, `SameSide(a,b,c,d)`. These conditions are directly obtained from diagrams and are never used as goals for proving problems. A metric condition is encoded either as a predicate proposition (e.g., `Perpendicular(x,a,b,c)`) or as an equation (e.g., `Angle(a,x,b) = π/2`).

Here are some exmaples of metric relations:

$$\texttt{Collinear(a,b,c)}$$
$$\texttt{Parallel(a,b,c,d)}$$
$$\texttt{Midpoint(a,b,c)}$$
$$\texttt{Perpendicular(a,b,c,d)}$$
$$\texttt{Congruent3(a,b,c,d,e,f)}$$
$$\texttt{Similar3(a,b,c,d,e,f)}$$

Here are some examples of diagrammatic relations:

$$\texttt{Between(a,b,c)}$$
$$\texttt{SameSide(a,b,c,d)}$$
$$\texttt{OppositeSide(a,b,c,d)}$$

The catalog of metric relations is intentionally extensible: new relations can be introduced by specifying their rules of introduction and elimination. For example, we define

$$\texttt{Square(a,b,c,d) := Rectangle(a,b,c,d)} \land \texttt{Length(a,b)=Length(a,d)}$$

to provide succinct definitions and proofs involving specific shapes.

We find it necessary to introduce diagrammatic relations in order to reason about angles in a human-like way. Two angles are considered equal if and only if they have the same cosine value. Consider the inscribed angle theorem (Figure 3a). If points $A, B, C, D$ lie on the same circle, then `Angle(A,C,B)` is either equal or supplementary to `Angle(A,D,B)`, depending on whether $A$ and $D$ lie on the same side or on opposite sides of line $AB$.

Without diagrammatic relations, systems like AlphaGeometry (Trinh et al., 2024) and SeedGeometry (Chen et al., 2025) cannot distinguish these two scenarios. Alternatively, they employ the *full-angle notation* (Chou et al., 1996), where two angles are considered equal if and only if they have the same sine value.

For instance, in Figure 3b, under the usual definition of angles:

$$\texttt{Angle(A,O,C) = Angle(C,O,A)}, \quad \texttt{Angle(A,O,C) + Angle(A,O,D) = π}$$

However, in full-angle notation:

$$\texttt{Angle(A,O,C) = Angle(A,O,D)}, \quad \texttt{Angle(A,O,C) = −Angle(C,O,A)}$$

This misalignment makes it impossible to faithfully translate problems between natural language and the formal language of AlphaGeometry (Trinh et al., 2024). In proving problems, the goal of establishing equality of two angles must instead be formulated as "equal or supplementary." This issue becomes more problematic for calculation problems, where the solution cannot uniquely determine the value of an angle.

Since diagrammatic inferences (Figure 3c) are almost always absent in human geometric literature (except in the context of formal logic), we directly extract them from diagrams as initial conditions. Incorporating diagrammatic inference rules into the deduction system would not only make

it more complex (requiring proof by contradiction and disjunctions), but would also render proofs unnecessarily verbose and less focused.

Unlike System-E (Avigad et al., 2009), which distinguishes three types of geometric objects—points, lines, and circles—we treat *points* as the only first-class citizens, and assume all points are implicitly connected. For example, our assertion

$$\texttt{Collinear(a,b,c)}$$

is equivalent to the System-E formulation:

$$\texttt{a,b,c: Point, l: Line, on(a,l), on(b,l), on(c,l)}$$

This simplification reduces the complexity of our rule-based deduction system and avoids trivial auxiliary constructions such as "Connect point `a` and point `b`."

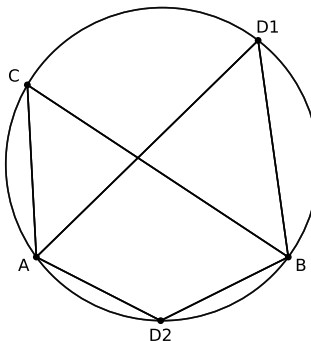 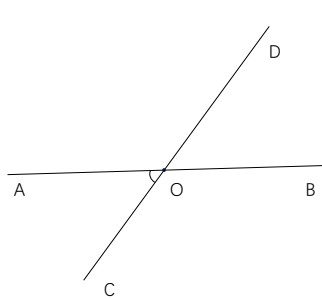 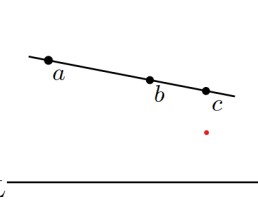

(a) The inscribed angle theorem: `Angle(A,C,B)` is equal to `Angle(A,D1,B)`, but supplementary to `Angle(A,D2,B)`.

(b) With the usual definition of angles: `Angle(A,O,C) = Angle(C,O,A)`, and `Angle(A,O,C)` is supplementary to `Angle(A,O,D)`.

(c) An example of a diagrammatic inference rule (from Avigad et al. (2009)): if `b` is between `a` and `c`, and `a` and `c` are on the same side of line $L$, then `a` and `b` are on the same side of $L$.

### A.2 PROBLEM CONSTRUCTION

In order to generate a problem, we first sample a list of construction rules and then sample a corresponding diagram. For calculation problems, we use parameterized construction rules. The diagram is fully determined by its construction rules and parameters, up to a global translation, reflection, and rotation. On the other hand, construction rules with unspecified degrees of freedom are also employed for generating proving problems, where the theorem to be proved usually holds for a range of diagrams rather than a specific one.

Consider the constructions

$$\texttt{a,b,c = construct\_triangle(); x = construct\_foot(a,b,c)}$$

illustrated in figures 4a and 4b. To sketch the diagram, the coordinates of points are sampled according to the construction rules. Some initial conditions directly follow from the construction, so we obtain `Not(Collinear(a,b,c))` and `Perpendicular(x,a,b,c)`. Additionally, diagrammatic relations obtained from the sample are added to the proving state, such as `Between(x,b,c)` and `OppositeSide(b,c,a,x)`. Given the same set of construction rules, the diagrammatic relations may differ from one sample to another. For example, if `Angle(a,b,c)` $> \pi/2$, we will instead obtain `Between(b,x,c)` and `SameSide(b,c,a,x)`.

### A.3 DEDUCTIVE DATABASE

After solving equations, we store all implied equivalence relationships in `union-find` structures. The list of points, the equivalence classes of angles, angle sums, lengths, and length ratios, together with each kind of predicates (such as `Perpendicular`, `Collinear`), is stored in an in-memory

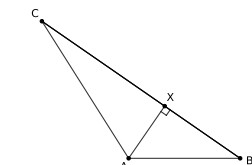
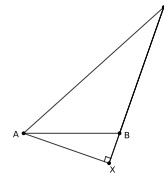
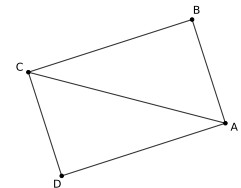

(a) A sampled figure with diagrammatic relations `Between(x,b,c)` and `OppositeSide(b,c,a,x)`.

(b) An alternative configuration with `Between(b,x,c)` and `SameSide(b,c,a,x)`.

(c) A diagram illustrating our algebraic system.

`SQL` database as tables. In this way, a conjunction of conditions is naturally encoded as a table join, and all applicable inference rules can be enumerated with an `SQL` query.

For example, the condition for `a` being the midpoint of `b` and `c` is:

```
Length(a,b) = Length(a,c), Collinear(a,b,c), Between(a,b,c)
```

which can be translated into an `SQL` query:

```
SELECT a.name AS a, b.name AS b, c.name AS c
FROM points a
JOIN points b
JOIN points c
JOIN length r0l
  ON ((r0l.p0 = a.name AND r0l.p1 = b.name) OR (r0l.p1 = a.name AND
r0l.p0 = b.name))
JOIN length r0r
  ON ((r0r.p0 = a.name AND r0r.p1 = c.name) OR (r0r.p1 = a.name AND
r0r.p0 = c.name))
  AND r0l.component = r0r.component
JOIN collinear r1
  ON ((a.name = r1.p0 AND b.name = r1.p1 AND c.name = r1.p2)
    OR (a.name = r1.p0 AND c.name = r1.p1 AND b.name = r1.p2)
    OR (b.name = r1.p0 AND a.name = r1.p1 AND c.name = r1.p2)
    OR (b.name = r1.p0 AND c.name = r1.p1 AND a.name = r1.p2)
    OR (c.name = r1.p0 AND a.name = r1.p1 AND b.name = r1.p2)
    OR (c.name = r1.p0 AND b.name = r1.p1 AND a.name = r1.p2))
JOIN between r2
  ON ((a.name = r2.p0 AND b.name = r2.p1 AND c.name = r2.p2)
    OR (a.name = r2.p0 AND c.name = r2.p1 AND b.name = r2.p2))
WHERE b.name < c.name;
```

Notice that we impose a lexical partial order on the variables to eliminate permutational redundancy. For example, `Midpoint(A,B,C)` is equivalent to `Midpoint(A,C,B)`, so the filter `b.name < c.name` removes the latter.

### A.4 ALGEBRAIC SYSTEM

During each iteration, our algebraic system solves the following three (log-linear) systems of equations using Gaussian elimination:

Linear equations of angles: $\texttt{Angle(A,B,C)} + \texttt{Angle(A,C,B)} + \texttt{Angle(B,A,C)} = \pi,$

Linear equations of lengths: $\texttt{Length(A, M)} + \texttt{Length(B,M)} = \texttt{Length(A,B)},$

Log-linear equations of lengths: $\dfrac{\texttt{Length(A,B)}}{\texttt{Length(P,Q)}} = \dfrac{\texttt{Length(B,C)}}{\texttt{Length(Q,R)}}.$

Although the linear and log-linear equations of lengths involve the same set of variables and could theoretically be solved together, doing so easily leads to high-order polynomials, making it impossible to find symbolic solutions. Instead, the three systems are solved independently.

Then, we substitute the solutions from each system into all equations, including the non-(log-)linear ones involving trigonometry or high-order polynomials. If the result contains no free variables, we have found the value of a geometric quantity. If the equation after substitution is (log-)linear, we add the equation back to the corresponding system. In this way, we allow message passing between the three (log-)linear systems without solving the entire non-linear system simultaneously.

Consider diagram A.4, where it is known that

$$\texttt{AD} + \texttt{AB} = 7, \quad \texttt{AD} - \texttt{AB} = 1.$$

The goal is finding $\texttt{AC}$. The set of linear equations of lengths can be summarized as the following matrix equation:

$$\begin{pmatrix} 1 & 1 & 0 & 0 \\ -1 & 1 & 0 & 0 \\ 1 & 0 & 0 & -1 \\ 0 & 1 & -1 & 0 \end{pmatrix} \begin{pmatrix} \texttt{AB} \\ \texttt{AD} \\ \texttt{BC} \\ \texttt{CD} \end{pmatrix} = \begin{pmatrix} 7 \\ 1 \\ 0 \\ 0 \end{pmatrix}$$

First, Gaussian elimination is applied to the system of linear equations to find $\texttt{AD} = 4$ and $\texttt{AB} = 3$. Using the Pythagorean theorem, we have

$$\texttt{AC}^2 = \texttt{AB}^2 + \texttt{AD}^2$$

Substituting the solutions from the linear system, we obtain

$$\texttt{AC}^2 = 25$$

After post-processing the solution and eliminating negative values for lengths, we obtain $\texttt{AC} = 5$.

The equation $\texttt{AC} = 5$ is both a linear and log-linear equation of lengths, so it is added back to both systems. The "sources" attribute is set as $AC^2 = AB^2 + AD^2, AB = 3, AD = 4$, which will be used during backtracking to generate the proof.

# B  MORE DETAILS OF EUCLID-OMNI

## B.1  EXAMPLES OF TEMPLATES AND PROMPTS FOR NATURAL LANGUAGE TRANSLATION

Following prior work (Huang et al., 2025), we construct multiple natural language templates for each construction rule (in the problem) and relation (in the solution) within our formalization. During transformation, one template is randomly sampled for instantiation. For example, the construction rule $\texttt{x = construct\_circumcenter(a,b,c)}$ can be expressed as:

```
x is the circumcenter of abc
x is the center of the circle passing through a, b, and c
the center of the circle through points a, b, and c is x
the point x is the circumcenter of triangle abc
```

Similarly, each relation can also be expressed through multiple templates. For example, the relation $\texttt{Perpendicular(a,b,c,d)}$ can be instantiated as:

```
line ab is perpendicular to line cd
line ab ⊥ line cd
line through a and b is perpendicular to the line through c and d
the lines formed by (a, b) and (c, d) are perpendicular
```

For LLM refinement, we first apply a problem prompt to transform the template-based problem and its goal into a more natural and fluent form. The prompt is defined as follows:

```
You are given a plane geometry problem:

Problem: <template-based problem>. Determine <template-based goal>.

Task:
```

```
- Rewrite the problem in clear, concise, and fluent language,
preserving the original meaning.
- Output ONLY the rewritten problem, with no explanations or extra
text.
```

Once the refined problem is obtained, we apply a solution prompt to convert the template-based solution into fluent text:

```
You are given a plane geometry problem and its corresponding
solution:

Problem:
<refined problem>

Solution:
<template-based solution>

Task:
- Rewrite the solution in clear, concise, and fluent language,
simplifying trivial or redundant steps.
- Step-wise formatting is optional. Use it only when it improves
clarity; otherwise, presenting the solution as a continuous
paragraph is acceptable.
- Output ONLY the rewritten solution, with the final answer inside
\boxed{} at the end.
- Do NOT include the problem statement, explanations, or extra
text.
```

We also support transforming formal problems and solutions into multiple-choice problems, as commonly used in calculation-style geometry datasets. The problem prompt for this setting is:

```
You are given a plane geometry problem and its corresponding
solution:

Problem: <template-based problem>. <template-based goal> = ( ).

Reference Answer (for correctness only):
<answer>

Task:
- Rewrite the problem in clear, concise, and fluent language,
preserving the original meaning.
- Convert the task into a multiple-choice question with exactly 4
options labeled A,B,C,D.
- Use the reference solution ONLY to determine the correct
numeric/choice answer.
- Create plausible distractors of comparable scale or magnitude to
the correct answer.
- Ensure EXACTLY ONE option is correct.
- Output ONLY the rewritten problem followed by the choices, with
no explanations or extra text.
- Do NOT include the solution, rationales, or extra text.

Output format:
<Rewritten problem statement>

Choices:
A: ...
B: ...
C: ...
D: ...
```

Finally, the solution prompt for the multiple-choice setting is:

```
You are given a plane geometry problem and its corresponding
solution:

Problem:
<refined multiple-choice problem>

Solution:
<template-based solution>

Task:
- Rewrite the solution in clear, concise, and fluent language,
simplifying trivial or redundant steps.
- Step-wise formatting is optional. Use it only when it improves
clarity; otherwise, presenting the solution as a continuous
paragraph is acceptable.
- Ensure the final choice label matches the provided solution's
final answer.
- Output ONLY the rewritten solution, with the final CHOICE LABEL
(e.g., A, B, C, or D) inside \boxed{} at the end.
- Do NOT include the problem statement, explanations, or extra
text.
```

## B.2 EXAMPLES OF SYNTHETIC INSTANCES

We present two examples of our generated instances for calculation problems: one in a final-answer format and the other in a multiple-choice format. The first example is shown below, with its corresponding diagram provided in Figure 5.

```
Problem:
ABCD is a square with side length 100. Point F lies on a circle
centered at D with a radius of 150. Line segment FB is
perpendicular to DB, and point G is the foot of the perpendicular
dropped from point B onto the line DF. Find the area of triangle
BDG.
```

```
Solution:
Let ABCD be a square with a side length of 100. Point F lies on a
circle centered at D with a radius of 150, and FB is perpendicular
to DB. Point G is the foot of the perpendicular from B to line DF.
\n\nSince triangle BDF and triangle BDG share ∠BDF and are both
right triangles, they are similar. Using the similarity ratio, we
have:\n\n\\[\n\\frac{BD}{DF} = \\frac{DG}{BD}\n\\]\n\nFrom the
square, \\( BD = 100\\) and \\( DF = 150 \\),
so:\n\n\\[\n\\frac{100}{150} = \\frac{DG}{100} \\implies DG =
\\frac{400}{3}.\n\\]\n\nSince \\( DF = DG + FG \\) and \\( DF =
150 \\), substituting \\( DG = \\frac{400}{3}
\\):\n\n\\[\n\\frac{400}{3} + FG = 150 \\implies FG =
\\frac{50}{3}.\n\\]\n\nTriangle BDG has a right angle at G. Its
area is given by:\n\n\\[\n\\text{Area} = \\frac{1}{2} \\times BG
\\times DG.\n\\]\n\nUsing the similarity ratio from triangles BDG
and FBG:\n\n\\[\n\\frac{BG}{DG} = \\frac{FG}{BG} \\implies BG^2 =
DG \\cdot FG.\n\\]\n\nSubstituting \\( DG = \\frac{400}{3} \\) and
\\( FG = \\frac{50}{3} \\):\n\n\\[\nBG^2 = \\frac{400}{3} \\cdot
\\frac{50}{3} = \\frac{20000}{9} \\implies BG =
\\sqrt{\\frac{20000}{9}} = \\frac{100\\sqrt{2}}{3}.\n\\]\n\nThe
area of triangle BDG is:\n\n\\[\n\\text{Area} = \\frac{1}{2}
\\times BG \\times DG = \\frac{1}{2} \\times
\\frac{100\\sqrt{2}}{3} \\times \\frac{400}{3} = \\frac{20000
\\sqrt{2}}{9}.\n\\]\n\n\\[\n\\boxed{\\frac{20000
\\sqrt{2}}{9}}\n\\]
```

The second example is shown below, with its corresponding diagram provided in Figure 6.

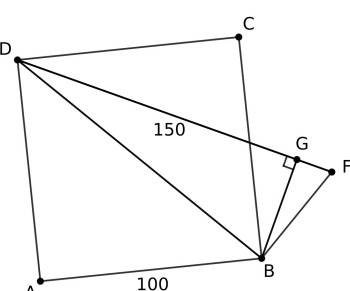

Figure 5: Diagram of the generated final-answer calculation problem.

```
Problem:
In triangle ABC, ∠ABC is a right angle, AB = 52, and BC = 32. D is
a point on the line passing through C and parallel to AB, such
that DB is perpendicular to AC. Find the area of triangle BCD.

Choices:
A: 2048/13
B: 4096/13
C: 5120/13
D: 1024/13
```

```
Solution:
Given that ∠ABC is a right angle, AB is perpendicular to BC. Since
AB is parallel to CD, CD is also perpendicular to BC, making ∠BCD
= 90°. The area of triangle BCD is given by \\(\\frac{1}{2} \\cdot
BC \\cdot CD\\).\n\nTriangles ABC and BCD are similar because ∠ABC
= ∠BCD (both 90°), and the other angles are congruent by the
geometric constraints. By the similarity ratio, \\(\\frac{AB}{BC}
= \\frac{BC}{CD}\\). Substituting the given side lengths \\(AB =
52\\) and \\(BC = 32\\), we solve for
\\(CD\\):\n\n\\[\n\\frac{52}{32} = \\frac{32}{CD} \\implies CD =
\\frac{32 \\cdot 32}{52} = \\frac{1024}{52} =
\\frac{256}{13}.\n\\]\n\nThe area of triangle BCD
is:\n\n\\[\n\\text{Area} = \\frac{1}{2} \\cdot BC \\cdot CD =
\\frac{1}{2} \\cdot 32 \\cdot \\frac{256}{13} =
\\frac{4096}{13}.\n\\]\n\n\\(\\boxed{B}\\)
```

We also provide two examples of synthetic problems that require auxiliary constructions. The first
example is presented below and the associated diagram is shown in Figure 7.

```
Problem:
a,b = construct_segment(), c = construct_on_circle(b,a),
c = construct_on_line(b,a), e = construct_on_bline(c,a),
f = construct_angle_bisector(b,c,e), f = construct_on_bline(e,a)
Goal:
Concyclic(a,c,e,f)
```

```
Auxiliary Constructions:
d = construct_midpoint(b,a),
h = construct_intersection_tt(f,a,e,b,c,d)
```

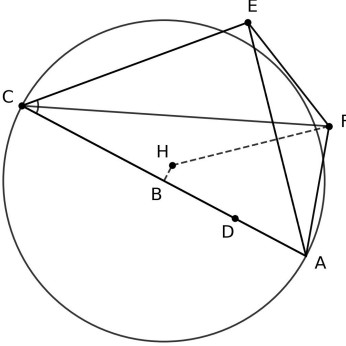

Figure 6: Diagram of the generated multiple-choice calculation problem.

Figure 7: Diagram of the synthetic problem with its auxiliary constructions.

The second example is presented below and the associated diagram is shown in Figure 8.

```
Problem:
a,b = construct_segment(), c = construct_on_dia(b,a),
d = construct_on_bline(c,a), d = construct_angle_bisector(c,b,a)
Goal:
Angle_a_b_c + Angle_a_d_c - 180
```

```
Auxiliary Constructions:
e = construct_on_dia(a,b), e = construct_angle_bisector(c,d,a)
```

## C    MORE DETAILS OF EXPERIMENTAL RESULTS

### C.1    EXAMPLES OF PRODUCED PROOFS

We compare the proofs generated by Euclidea against those produced by AlphaGeometry (Trinh et al., 2024), Newclid (Sicca et al., 2024), and PyEuclid (Li et al., 2025). For this evaluation, we

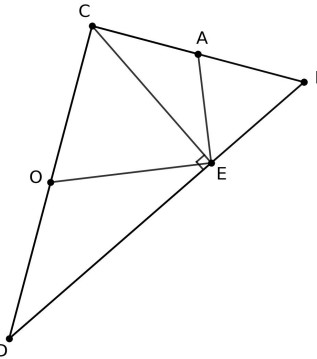

Figure 8: Diagram of the synthetic problem with its auxiliary constructions.

randomly selected two problems: one from JGEX-AG-231 (Trinh et al., 2024) and another from IMO-AG-30 (Trinh et al., 2024).

The natural language formulation of the first problem is given below, and the corresponding diagram is shown in Figure 9.

```
In triangle ECD, ∠E is a right angle. O is the midpoint of side
DC. Line AC is perpendicular to side DC, and AE is perpendicular
to EO. Line CA intersects at a point F, and line DE also passes
through point F. Prove that AE is equal to AF.
```

Figure 9: Diagram of the geometry problem selected from JGEX-AG-231.

The proof produced by Euclidea for this problem is shown below:

```
Solution:
1. Collinear(d,e,f) => Angle_e_d_o - Angle_f_d_o & Angle_a_f_d -
Angle_a_f_e & Angle_d_e_o + Angle_f_e_o - 180
2. Collinear(c,d,o) => Parallel(c,d,d,o)
3. Perpendicular(a,c,c,d) & Parallel(c,d,d,o) =>
Perpendicular(a,c,d,o)
4. Collinear(a,c,f) => Parallel(a,c,a,f)
```

```
5. Perpendicular(a,c,d,o) & Parallel(a,c,a,f) =>
Perpendicular(a,f,d,o)
6. Perpendicular(a,f,d,o) => Angle_a_f_d + Angle_f_d_o - 90
7. Angle_e_d_o - Angle_f_d_o & Angle_a_f_d - Angle_a_f_e &
Angle_a_f_d + Angle_f_d_o - 90 => Angle_a_f_e + Angle_e_d_o - 90
8. Perpendicular(a,e,e,o) => Angle_a_e_o - 90
9. Angle_a_e_f + Angle_a_e_o - Angle_f_e_o & Angle_d_e_o +
Angle_f_e_o - 180 & Angle_a_e_o - 90 => Angle_a_e_f + Angle_d_e_o
- 90
10. Midpoint(o,c,d) & Perpendicular(c,e,d,e) => Length_d_o -
Length_e_o
11. Length_d_o - Length_e_o => Angle_d_e_o - Angle_e_d_o
12. Angle_a_f_e + Angle_e_d_o - 90 & Angle_a_e_f + Angle_d_e_o -
90 & Angle_d_e_o - Angle_e_d_o => Angle_a_e_f - Angle_a_f_e
13. Angle_a_e_f - Angle_a_f_e => Length_a_e - Length_a_f
```

The proof produced by AlphaGeometry for this problem is shown below:

```
 * Proof steps:
001. C,O,D are collinear [01] & CD ⊥ AC [03] ⇒  CO ⊥ CA [07]
002. CO ⊥ CA [07] & AE ⊥ EO [04] ⇒  ∠COE = ∠CAE [08]
003. C,A,F are collinear [05] & C,O,D are collinear [01] & ∠COE =
∠CAE [08] ⇒  ∠FAE = ∠COE [09]
004. D,F,E are collinear [06] & DE ⊥ CE [00] ⇒  FD ⊥ CE [10]
005. AE ⊥ EO [04] & FD ⊥ CE [10] ⇒  ∠(AE-FD) = ∠OEC [11]
006. D,F,E are collinear [06] & ∠(AE-FD) = ∠OEC [11] ⇒  ∠FEA = ∠
CEO [12]
007. ∠FAE = ∠COE [09] & ∠FEA = ∠CEO [12] (Similar Triangles)⇒
OC:OE = AF:AE [13]
008. C,O,D are collinear [01] & OD = OC [02] ⇒  O is midpoint of
DC [14]
009. CE ⊥ DE [00] & O is midpoint of DC [14] ⇒  CO = EO [15]
010. OC:OE = AF:AE [13] & CO = EO [15] ⇒  AF = AE
```

The proof produced by Newclid for this problem is shown below:

```
# Proof:
000. | O is the midpoint of CD [C0], CE ⊥ DE [C1] =(r19 Hypotenuse
is diameter)> CO = EO [0]
001. | CO = EO [0] =(r13 Isosceles triangle equal angles)> ∠
(CE,CO) = ∠(EO,CE) [1]
002. | A, C, F are collinear [C2], A ≠ C [N0], A ≠ F [N1], C ≠ F
[N2] =(r82 Parallel from collinear)> AF ∥ AC [2]
003. | O is the midpoint of CD [C0] =(r56 Properties of midpoint
(coll))> C, D, O are collinear [3]
004. | C, D, O are collinear [3], C ≠ D [N3], C ≠ O [N4], D ≠ O
[N5] =(r82 Parallel from collinear)> CO ∥ CD [4]
005. | D, E, F are collinear [C3], D ≠ E [N6], D ≠ F [N7], E ≠ F
[N8] =(r82 Parallel from collinear)> DE ∥ EF [5]
006. | ∠(CE,CO) = ∠(EO,CE) [1], AF ∥ AC [2], CO ∥ CD [4], DE ∥ EF
[5], AC ⊥ CD [C4], AE ⊥ EO [C5], CE ⊥ DE [C1] =(AR Deduction)>
∠(AE,EF) = ∠(EF,AF) [6]
007. | ∠(AE,EF) = ∠(EF,AF) [6], A, E, F are not collinear [N9]
=(r14 Equal base angles imply isosceles)> AF = AE [7]
```

The proof produced by PyEuclid for this problem is shown below:

```
 * Proof steps:
001. Length_c_o - Length_d_o &
Collinear(c,d,o) &
Between(o,c,d) ⇒ -Length_c_d/2 + Length_d_o
002. Angle_c_d_e - Angle_f_d_o &
Angle_c_d_f - Angle_f_d_o ⇒ Angle_c_d_e - Angle_c_d_f
003. Perpendicular(a,c,c,d) &
```

```
Parallel(a,c,c,f) ⇒ Angle_d_c_f - pi/2
004. Angle_d_c_f - pi/2(3) &
Angle_c_e_d - pi/2 ⇒ Angle_c_e_d - Angle_d_c_f
005. Not(Collinear(c,d,e)) &
Angle_c_d_e - Angle_c_d_f(2) &
Angle_c_e_d - Angle_d_c_f(4) ⇒ Length_c_d/Length_d_f -
Length_d_e/Length_c_d
006. Perpendicular(c,e,d,e) &
Parallel(d,e,e,f) ⇒ Angle_c_e_f - pi/2
007. Angle_c_e_d - pi/2 &
Angle_c_e_f - pi/2(6) ⇒ -Angle_c_e_d + Angle_c_e_f
008. Angle_c_e_d - pi/2 &
Angle_c_d_e - Angle_f_d_o &
Angle_c_d_e + Angle_c_e_d + Angle_d_c_e - pi ⇒ Angle_d_c_e +
Angle_f_d_o - pi/2
009. Angle_c_d_f - Angle_f_d_o &
Angle_c_f_d - Angle_c_f_e &
Angle_c_d_f + Angle_c_f_d + Angle_d_c_f - pi &
Angle_d_c_f - pi/2 ⇒ Angle_c_f_e + Angle_f_d_o - pi/2
010. Angle_d_c_e + Angle_f_d_o - pi/2(8) &
Angle_c_f_e + Angle_f_d_o - pi/2(9) ⇒ Angle_c_f_e - Angle_d_c_e
011. Not(Collinear(c,e,f)) &
-Angle_c_e_d + Angle_c_e_f(7) &
Angle_c_f_e - Angle_d_c_e(10) ⇒ Length_c_e/Length_d_e -
Length_e_f/Length_c_e
012. Angle_a_e_o - pi/2 &
Angle_a_f_e - Angle_c_f_d &
-Angle_c_d_f + Angle_f_d_o &
-Angle_a_e_f - Angle_a_e_o + Angle_f_e_o &
Angle_a_e_f + Angle_a_f_e + Angle_e_a_f - pi &
Angle_c_d_f + Angle_c_f_d + Angle_d_c_f - pi &
Angle_d_c_f - pi/2 ⇒ Angle_e_a_f - Angle_f_d_o + Angle_f_e_o - pi
013. -Angle_e_d_o + Angle_f_d_o &
Angle_d_e_o + Angle_d_o_e + Angle_e_d_o - pi &
Angle_c_o_e + Angle_d_o_e - pi &
Angle_d_e_o + Angle_f_e_o - pi ⇒ Angle_c_o_e - Angle_f_d_o +
Angle_f_e_o - pi
014. Angle_e_a_f - Angle_f_d_o + Angle_f_e_o - pi(12) &
Angle_c_o_e - Angle_f_d_o + Angle_f_e_o - pi(13) ⇒ -Angle_c_o_e +
Angle_e_a_f
015. -Angle_c_e_f - Angle_c_e_o + Angle_f_e_o &
Angle_c_e_f - pi/2 ⇒ Angle_c_e_o - Angle_f_e_o + pi/2
016. Angle_a_e_o - pi/2 &
-Angle_a_e_f - Angle_a_e_o + Angle_f_e_o ⇒ Angle_a_e_f -
Angle_f_e_o + pi/2
017. Angle_c_e_o - Angle_f_e_o + pi/2(15) &
Angle_a_e_f - Angle_f_e_o + pi/2(16) ⇒ Angle_a_e_f - Angle_c_e_o
018. Not(Collinear(a,e,f)) &
-Angle_c_o_e + Angle_e_a_f(14) &
Angle_a_e_f - Angle_c_e_o(17) ⇒ Length_a_f/Length_c_o -
Length_e_f/Length_c_e
019. -Length_c_o + Length_d_o &
-Length_c_d/2 + Length_d_o(1) &
Length_c_d/Length_d_f - Length_d_e/Length_c_d(5) &
Length_c_e/Length_d_e - Length_e_f/Length_c_e(11) &
Length_a_f/Length_c_o - Length_e_f/Length_c_e(18) ⇒ Length_a_f -
sqrt(Length_d_f)*sqrt(Length_e_f)/2
020. -Angle_a_f_e + Angle_c_f_d &
Angle_c_d_f - Angle_f_d_o &
Angle_c_d_f + Angle_c_f_d + Angle_d_c_f - pi &
Angle_d_c_f - pi/2 ⇒ Angle_a_f_e + Angle_f_d_o - pi/2
021. Angle_c_e_d - pi/2 &
Angle_c_d_e - Angle_f_d_o &
Angle_d_c_e - Angle_e_c_o &
```

```
Angle_c_d_e + Angle_c_e_d + Angle_d_c_e - pi ⇒ Angle_e_c_o +
Angle_f_d_o - pi/2
022. Angle_a_f_e + Angle_f_d_o - pi/2(20) &
Angle_e_c_o + Angle_f_d_o - pi/2(21) ⇒ Angle_a_f_e - Angle_e_c_o
023. Not(Collinear(a,e,f)) &
Angle_a_e_f - Angle_c_e_o(17) &
Angle_a_f_e - Angle_e_c_o(22) ⇒ Length_a_e/Length_e_o -
Length_e_f/Length_c_e
024. Angle_e_d_o - Angle_f_d_o &
Angle_d_e_o + Angle_d_o_e + Angle_e_d_o - pi &
Angle_d_e_o + Angle_f_e_o - pi ⇒ Angle_d_o_e + Angle_f_d_o -
Angle_f_e_o
025. Angle_a_e_o - pi/2 &
-Angle_a_f_e + Angle_c_f_d &
Angle_c_d_f - Angle_f_d_o &
-Angle_a_e_f - Angle_a_e_o + Angle_f_e_o &
Angle_a_e_f + Angle_a_f_e + Angle_e_a_f - pi &
Angle_c_d_f + Angle_c_f_d + Angle_d_c_f - pi &
Angle_c_a_e + Angle_e_a_f - pi &
Angle_d_c_f - pi/2 ⇒ Angle_c_a_e + Angle_f_d_o - Angle_f_e_o
026. Angle_d_o_e + Angle_f_d_o - Angle_f_e_o(24) &
Angle_c_a_e + Angle_f_d_o - Angle_f_e_o(25) ⇒ Angle_c_a_e -
Angle_d_o_e
027. Angle_a_c_d - pi/2 &
Angle_c_e_d - pi/2 &
-Angle_c_d_e + Angle_f_d_o &
Angle_a_c_d - Angle_a_c_e - Angle_d_c_e &
Angle_c_d_e + Angle_c_e_d + Angle_d_c_e - pi ⇒ Angle_a_c_e -
Angle_f_d_o
028. Angle_a_c_e - Angle_f_d_o(27) &
Angle_e_d_o - Angle_f_d_o ⇒ Angle_a_c_e - Angle_e_d_o
029. Not(Collinear(a,c,e)) &
Angle_c_a_e - Angle_d_o_e(26) &
Angle_a_c_e - Angle_e_d_o(28) ⇒ Length_a_e/Length_e_o -
Length_c_e/Length_d_e
030. Length_c_o - Length_d_o &
Collinear(c,d,o) &
Between(o,c,d) &
Perpendicular(c,e,d,e) ⇒ Length_c_o - Length_e_o
031. -Length_c_d/2 + Length_d_o(1) &
Length_a_e/Length_e_o - Length_e_f/Length_c_e(23) &
Length_c_d/Length_d_f - Length_d_e/Length_c_d(5) &
Length_a_e/Length_e_o - Length_c_e/Length_d_e(29) &
Length_d_o/Length_e_o - Length_e_o/Length_d_o ⇒ Length_a_e -
sqrt(Length_d_f)*sqrt(Length_e_f)/2
032. Length_a_f - sqrt(Length_d_f)*sqrt(Length_e_f)/2(19) &
Length_a_e - sqrt(Length_d_f)*sqrt(Length_e_f)/2(31) ⇒ Length_a_e
- Length_a_f
```

For the second problem, the natural language formulation is presented below, together with its corresponding diagram in Figure 10.

```
Let BC be a line segment, and O be the midpoint of BC. Point A
lies on a circle with center O and radius OB. Point D lies on the
perpendicular bisector of line segment AB and is located on the
same circle with center O and radius OB. Point E is located on the
perpendicular bisector of line segment OA and also lies on the
circle with center O and radius OB. Similarly, point F is located
on the perpendicular bisector of line segment OA and lies on the
circle with center O and radius OB. Line JO is parallel to AD, and
point J lies on line AC. Prove that ∠ECJ = ∠FCJ.
```

The proof produced by Euclidea for this problem is shown below:

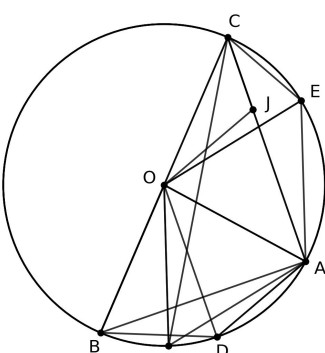

Figure 10: Diagram of the geometry problem selected from IMO-AG-30.

```
Solution:
1. Length_b_o - Length_c_o & Length_a_o - Length_b_o => Length_a_o
- Length_c_o
2. Length_b_o - Length_f_o & Length_b_o - Length_c_o => Length_c_o
- Length_f_o
3. -Length_a_o + Length_c_o & Length_c_o - Length_f_o =>
Angle_a_c_f - Angle_a_o_f/2
4. Collinear(a,c,j) => Angle_a_c_f - Angle_f_c_j & Angle_a_c_e -
Angle_e_c_j
5. Length_b_o - Length_f_o & Length_b_o - Length_e_o => Length_e_o
- Length_f_o
6. Length_a_e - Length_e_o & Length_e_o - Length_f_o & -Length_a_f
+ Length_f_o => Rhombus(a,e,o,f)
7. Rhombus(a,e,o,f) => Angle_a_o_e - Angle_a_o_f
8. Length_b_o - Length_c_o & Length_b_o - Length_e_o => Length_c_o
- Length_e_o
9. -Length_a_o + Length_c_o & Length_c_o - Length_e_o =>
Angle_a_c_e - Angle_a_o_e/2
10. Angle_a_c_f - Angle_a_o_f/2 & Angle_a_c_f - Angle_f_c_j &
Angle_a_o_e - Angle_a_o_f & Angle_a_c_e - Angle_a_o_e/2 &
Angle_a_c_e - Angle_e_c_j => Angle_e_c_j - Angle_f_c_j
```

The proof produced by AlphaGeometry for this problem is shown below:

```
 * Proof steps:
001. OE = OB [03] & OF = OB [05] & OA = OB [01] & OD = OB [02] &
OB = OC [00] ⇒  E,A,F,C are concyclic [08]
002. E,A,F,C are concyclic [08] ⇒  ∠AEF = ∠ACF [09]
003. E,A,F,C are concyclic [08] ⇒  ∠EFA = ∠ECA [10]
004. EO = EA [04] & OE = OB [03] & OF = OB [05] & FO = FA [06] ⇒
AF = AE [11]
005. AF = AE [11] ⇒  ∠EFA = ∠AEF [12]
006. J,A,C are collinear [07] & ∠AEF = ∠ACF [09] & ∠EFA = ∠AEF
[12] & ∠EFA = ∠ECA [10] ⇒  ∠ECJ = ∠JCF
```

The proof produced by Newclid for this problem is shown below:

```
# Proof:
000. | O is the midpoint of BC [C1] =(r51 Midpoint splits in two)>
BC:BO = 2/1 [0]
```

```
001. | O is the midpoint of BC [C1] =(r51 Midpoint splits in two)>
BC:CO = 2/1 [1]
002. | AO = BO [C0], BC:BO = 2/1 [0], BC:CO = 2/1 [1] =(AR
Deduction)> CO = AO [2]
003. | CO = AO [2] =(r13 Isosceles triangle equal angles)>
∠(AC,AO) = ∠(CO,AC) [3]
004. | ∠(AE,AO) = ∠(AO,EO) [C3], ∠(AF,AO) = ∠(AO,FO) [C2] =(AR
Deduction)> ∠(AE,AF) = ∠(FO,EO) [4]
005. | AE = EO [C4], EO = BO [C5], FO = AF [C6], FO = BO [C7] =(AR
Deduction)> AE:AF = FO:EO [5]
006. | ∠(AE,AF) = ∠(FO,EO) [4], AE:AF = FO:EO [5], △AEF has the
same orientation as △EOF [N0] =(r62 SAS Similarity of triangles
(Direct))> △AEF ≅ △OFE [6]
007. | △AEF has the same orientation as △EOF [N0], △AEF ≅ △OFE
[6] =(r52 Properties of similar triangles (Direct))> ∠(AF,EF) =
∠(EO,EF) [7]
008. | EO = BO [C5], BC:BO = 2/1 [0], BC:CO = 2/1 [1] =(AR
Deduction)> EO = CO [8]
009. | EO = CO [8] =(r13 Isosceles triangle equal angles)>
∠(CE,CO) = ∠(EO,CE) [9]
010. | FO = BO [C7], BC:BO = 2/1 [0], BC:CO = 2/1 [1] =(AR
Deduction)> CO = FO [10]
011. | CO = FO [10] =(r13 Isosceles triangle equal angles)>
∠(CF,CO) = ∠(FO,CF) [11]
012. | A, C, J are collinear [C8], A ≠ C [N1], A ≠ J [N2], C ≠ J
[N3] =(r82 Parallel from collinear)> CJ ∥ AC [12]
013. | ∠(AC,AO) = ∠(CO,AC) [3], ∠(AF,AO) = ∠(AO,FO) [C2],
∠(AF,EF) = ∠(EO,EF) [7], ∠(CE,CO) = ∠(EO,CE) [9], ∠(CF,CO) =
∠(FO,CF) [11], CJ ∥ AC [12] =(AR Deduction)> ∠(CE,CJ) = ∠(CJ,CF)
[13]
```

The proof produced by PyEuclid for this problem is shown below:

```
* Proof steps:
001. Length_a_o - Length_b_o &
-Length_b_o + Length_f_o ⇒ Length_a_o - Length_f_o
002. -Length_a_e + Length_e_o &
-Length_b_o + Length_e_o &
-Length_b_o + Length_f_o ⇒ Length_a_e - Length_f_o
003. Length_a_o - Length_f_o(1) &
Length_a_e - Length_f_o(2) ⇒ Length_a_e - Length_a_o
004. Not(Collinear(a,e,o)) &
Length_a_e - Length_a_o(3) ⇒ Angle_a_e_o - Angle_a_o_e
005. -Length_b_o + Length_e_o &
-Length_b_o + Length_f_o ⇒ Length_e_o - Length_f_o
006. Not(Collinear(a,e,o)) &
Length_a_e - Length_e_o ⇒ -Angle_a_o_e + Angle_e_a_o
007. Length_b_o - Length_c_o &
-Length_b_o + Length_f_o ⇒ Length_c_o - Length_f_o
008. Length_a_o - Length_f_o(1) &
Length_c_o - Length_f_o(7) ⇒ Length_a_o - Length_c_o
009. Length_e_o - Length_f_o(5) &
Length_c_o - Length_f_o(7) ⇒ Length_c_o - Length_e_o
010. SameSide(c,o,a,e) &
Length_a_o - Length_c_o(8) &
Length_c_o - Length_e_o(9) ⇒ Angle_a_c_e - Angle_a_o_e/2
011. Angle_a_c_e - Angle_e_c_j &
Angle_a_e_o + Angle_a_o_e + Angle_e_a_o - pi &
Angle_a_e_o - Angle_a_o_e(4) &
-Angle_a_o_e + Angle_e_a_o(6) &
Angle_a_c_e - Angle_a_o_e/2(10) ⇒ Angle_e_c_j - pi/6
012. Length_a_o - Length_f_o(1) &
Length_a_f - Length_f_o ⇒ Length_a_f - Length_a_o
013. Not(Collinear(a,f,o)) &
Length_a_f - Length_a_o(12) ⇒ Angle_a_f_o - Angle_a_o_f
```

```
014. Not(Collinear(a,f,o)) &
Length_a_f - Length_f_o ⇒ -Angle_a_o_f + Angle_f_a_o
015. SameSide(c,o,a,f) &
Length_a_o - Length_c_o(8) &
Length_c_o - Length_f_o ⇒ Angle_a_c_f - Angle_a_o_f/2
016. Angle_a_c_f - Angle_f_c_j &
Angle_a_f_o + Angle_a_o_f + Angle_f_a_o - pi &
Angle_a_f_o - Angle_a_o_f(13) &
-Angle_a_o_f + Angle_f_a_o(14) &
Angle_a_c_f - Angle_a_o_f/2(15) ⇒ Angle_f_c_j - pi/6
017. Angle_e_c_j - pi/6(11) &
Angle_f_c_j - pi/6(16) ⇒ Angle_e_c_j - Angle_f_c_j
```

It is important to note that proofs generated by different systems can vary substantially in both style and strategy. In contrast to Euclidea, AlphaGeometry and Newclid rely on full-angle formalization (Chou et al., 1996), which often fails to distinguish an angle from its supplement and may therefore yield "incorrect" angle relations compared to human reasoning (highlighted in red). Furthermore, these systems represent angles using pairs of lines (shown in blue), introducing additional ambiguity. By comparison, the proofs generated by Euclidea faithfully capture the angle relations shown in the diagram, yielding a representation that is more human-like. Compared to PyEuclid, Euclidea produces proofs that are about twice as compact and avoid redundant equations, making them significantly clearer and easier for humans to read.

### C.2 EXAMPLES OF GENERATED SOLUTIONS ON CALCULATION PROBLEMS

We provide two samples from GeoQA (Chen et al., 2021a) and Geometry3K (Lu et al., 2021) datasets respectively and compare the solutions given by our finetuned model with the one given by base Qwen 2.5 VL model to illustrate efficacy of training with the synthetic dataset and the capability of our model in solving the informal geometry problems.

The natural language description of the problem from GeoQA dataset is given below, and the corresponding diagram is shown in Figure 11. The example demonstrate that our model is able to understand diagrammatic relationship correctly via finetunig.

```
As shown in the figure, it is known that BA is the tangent of
circle O, and connect OB to intersect circle O at point C. If
angle B = 45.0 and the length of AB is 2.0, then the length of BC
is
()\nChoices:\nA:2\sqrt{2}-1\nB:\sqrt{2}\nC:2\sqrt{2}-2\nD:2-\sqrt{2}
```

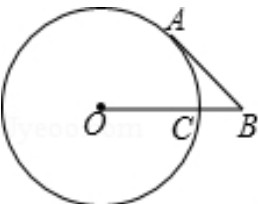

Figure 11: Diagram of the problem selected from GeoQA dataset.

The solution produced by our model for this problem is shown below:

```
Since BA is a tangent line to circle O, we have OA perpendicular
to AB. Given that angle B is 45 degrees, we can find that OA
equals AB, which is 2. In right triangle AOB, using the
Pythagorean theorem, we can find OB as \sqrt{OA^2+AB^2}, which
equals 2\sqrt{2}. Therefore, BC equals OB minus OC, which is
2\sqrt{2}-2. Hence, the answer is \\boxed{C}
```

The solution produced by the base model for this problem is shown below:

```
Since AB is a tangent line to circle O, we have OA perpendicular
AB. In right triangle AOB, we know that angle B is 45\u00b0 and AB
```

```
is 2. Therefore, OA = OB = AB * sin(45°) = \sqrt{2}. Hence, OC = OA
= \sqrt{2}. Thus, BC = OB - OC = (\sqrt{2}) - (\sqrt{2})/2 =
(\sqrt{2})/2. Therefore, the answer is option D.\nAnswer:D
```

The natural language description of the problem from Geometry3K dataset is given below, and the corresponding diagram is shown in Figure 12. The example shows that our model learns the correct calculation steps from the algebraic system of the symbolic solver.

```
Find $m\\angle R$\nChoices:\nA:58\nB:60\nC:122\nD:124
```

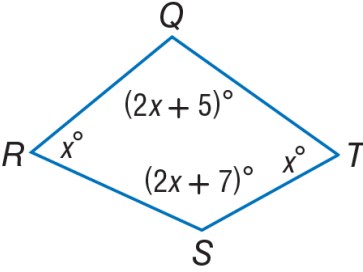

Figure 12: Diagram of the problem selected from Geometry3K dataset.

```
To find the measure of angle \\( \\angle R \\), we need to use the
fact that the sum of the interior angles of a quadrilateral is
always 360 degrees.\n\nGiven the angles in the quadrilateral:\n-
\\( \\angle Q = (2x + 5)^\\circ \\)\n- \\( \\angle R = x^\\circ
\\)\n- \\( \\angle S = (2x + 7)^\\circ \\)\n- \\( \\angle T =
x^\\circ \\)\n\nWe can set up the equation for the sum of the
interior angles:\n\\[\n(2x + 5) + x + (2x + 7) + x =
360\n\\]\n\nCombine like terms:\n\\[\n2x + 5 + x + 2x + 7 + x =
360\n\\]\n\\[\n6x + 12 = 360\n\\]\n\nSubtract 12 from both
sides:\n\\[\n6x = 348\n\\]\n\nDivide by 6:\n\\[\nx =
58\n\\]\n\nNow, substitute \\( x = 58 \\) back into the expression
for \\( \\angle R \\):\n\\[\n\\angle R = x^\\circ =
58^\\circ\n\\]\n\nThus, the measure of \\( \\angle R \\)
is:\n\\[\n\\boxed{58}\n\\]
```

```
To find the measure of angle \\( \\angle R \\) in the given
quadrilateral, we need to use the fact that the sum of the
interior angles of a quadrilateral is always \\( 360^\\circ
\\).\n\nThe given angles are:\n- \\( \\angle Q = (2x + 5)^\\circ
\\)\n- \\( \\angle S = (2x + 7)^\\circ \\)\n- \\( \\angle T =
x^\\circ \\)\n- \\( \\angle R = x^\\circ \\)\n\nWe can set up the
equation for the sum of the interior angles:\n\n\\[\n(2x + 5) +
(2x + 7) + x + x = 360\n\\]\n\nCombine like terms:\n\n\\[\n2x + 5
+ 2x + 7 + x + x = 360\n\\]\n\n\\[\n4x + 12 = 360\n\\]\n\nSubtract
12 from both sides:\n\n\\[\n4x = 348\n\\]\n\nDivide by
4:\n\n\\[\nx = 87\n\\]\n\nNow, substitute \\( x = 87 \\) back into
the expression for \\( \\angle R \\):\n\n\\[\n\\angle R = x^\\circ
= 87^\\circ\n\\]\n\nThus, the measure of \\( \\angle R \\)
is:\n\n\\[\n\\boxed{87}\n\\]
```

## C.3 EXAMPLES OF GENERATED AUXILIARY CONSTRUCTIONS ON PROVING PROBLEMS

We compared the auxiliary constructions identified by AlphaGeometry (Trinh et al., 2024) with those generated by our approach. Notably, our predicted constructions often differ from those of Alpha-Geometry, highlighting that multiple valid auxiliary strategies can achieve the same goal. Moreover, our method sometimes requires fewer auxiliary constructions than AlphaGeometry. To illustrate these differences, we present two examples from the IMO-AG-30 dataset (Trinh et al., 2024).

The natural language formulation of the first problem is given below, and the corresponding diagram is shown in Figure 13.

```
In triangle ABC, point O is the midpoint of side BC. Point M lies
on the circle centered at O with radius OB, and M is located on
line AB. Point N lies on the circle centered at O with radius OB,
and N is located on line AC. Point R lies on the angle bisector of
∠BAC, such that ∠MOR = ∠RON. Let O₁ be the center of the circle
passing through points B, M, and R, and O₂ be the circumcenter of
triangle CNR. Point P lies on the circle centered at O₁ with radius
O₁R, and P also lies on the circle centered at O₂ with radius O₂R.
Prove that the points B, C, and P are collinear.
```

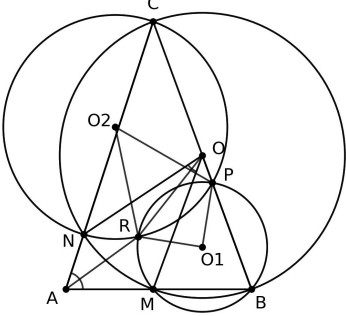

Figure 13: Diagram of the first geometry problem selected from IMO-AG-30.

The auxiliary constructions predicted by our LLLM for this problem is shown below:

```
Construct point K as the circumcenter of triangle AMN.
```

The auxiliary constructions predicted by AlphaGeometry for this problem is shown below:

```
Construct point K such that KM = KN.
Construct point L as the intersection of circles (K, A) and (O, A).
```

The natural language formulation of the second problem is given below, and the corresponding diagram is shown in Figure 14.

```
In triangle ABC, let H be the orthocenter. Point F lies on the
line HA and also on the line BC. Let M be the midpoint of segment
BC. Let O be the circumcenter of triangle ABC, which is the center
of the circle passing through points A, B, and C. Triangle QAH is
a right triangle with a 90-degree angle at Q, where Q lies on the
circle centered at O with radius OA. Similarly, triangle KHQ is a
right triangle with a 90-degree angle at K, where K also lies on
the circle centered at O with radius OA. Let O₁ be the circumcenter
of triangle KQH, and let O₂ be the circumcenter of triangle FKM.
Prove that points K, O₁, and O₂ are collinear.
```

The auxiliary constructions predicted by our LLLM for this problem is shown below:

```
Construct point p as the intersection of cicle (O, A) and Line (H,
Q).
```

Figure 14: Diagram of the second geometry problem selected from IMO-AG-30.

The auxiliary constructions predicted by AlphaGeometry for this problem is shown below:

```
Construct point X as the midpoint of CH.
Construct point Y as the midpoint of KM.
Construct point Z as the midpoint of BH.
```

## D   THE USE OF LLMS

LLMs were mainly used for minor language editing. Specifically, we employed them to polish the phrasing of individual sentences or short paragraphs in order to improve fluency and readability.

