# OpenReview forum: "Euclid-Omni: A Unified Neuro-Symbolic Framework for Geometry Problem Solving"
_ICLR.cc/2026/Conference — Submitted to ICLR 2026_

### Official Review · Reviewer_qtUi · 2025-10-16

**Soundness:** 3
**Presentation:** 2
**Contribution:** 2
**Rating:** 2
**Confidence:** 4

**Summary:**

This paper aims to address the limitations of existing AI-based geometry problem solvers, which are often narrowly scoped to specific problem types (e.g., theorem proving) and require enormous computational resources for training. The authors first propose Euclidea, which is a symbolic solver that integrates a deductive database with an algebraic engine to generate human-readable solutions. Then, they propose Euclid-Omni, which is a data generation pipeline that synthesizes problems, renders diagrams, and translates symbolic solutions into natural language for VLM training. Experiments are conducted to evaluate the symbolic solver and the models trained on the synthetic data. The results demonstrate that Euclidea can solve more problems than prior symbolic systems , and the trained models achieve competitive performance on various benchmarks while using orders of magnitude less training data.

**Strengths:**

1. The paper provides many examples in the main text and appendices, which are helpful for understanding the implementation details.

2. The proposed data generation pipeline is a good contribution to the field. It addresses a well-known bottleneck of data scarcity in geometric reasoning.

**Weaknesses:**

1. The proposed framework, Euclidea, appears to be heavily reliant on manually defined components. It is driven by a Deductive Database of pre-defined theorems and an Algebraic System that categorizes all equations into only four pre-defined types. I am doubtful that this manually curated set of theorems and equation types can cover the vast and complex landscape of real-world geometric reasoning. The completeness and generalization of this symbolic system are potential issues that are not sufficiently discussed.

2. The paper's description of its contributions, particularly the term "neuro-symbolic framework," is confusing. The abstract introduces "Euclid-Omni, a unified neuro-symbolic framework that combines a formal geometry system with Large (Vision)-Language Models (LLMs and VLMs) to address... problems". This phrasing is misleading for two reasons. First, it suggests that Euclid-Omni is the reasoning framework itself. However, after reading the paper, my understanding is that Euclidea is the purely symbolic reasoning system, while Euclid-Omni is the name for the data generation pipeline, which uses both the symbolic Euclidea and neural LLMs (for translation). Second, the abstract's phrasing implies that the problem-solving process is a deep integration of symbolic logic and neural reasoning. In reality, the core reasoning is performed entirely by the symbolic Euclidea system. The "neuro" components serve auxiliary, non-reasoning roles, such as translating solutions into "human-readable reasoning traces". This discrepancy severely misrepresents the paper's core contribution.

3. The experimental improvements in reasoning performance over existing baselines are not overwhelmingly significant, and furthermore, many key explanations and analyses of the results are missing (please refer to Q1-Q2 below).

**Questions:**

1. In the experiments, the authors use different settings for the two main tasks. In Section 4.2 (calculation), 20K training samples and the Qwen2.5-VL model are used. In Section 4.3 (proving), 100K samples and the Qwen2.5-Math-7B model are used. Could the authors explain the rationale behind these different data sizes and backbone models? Given the paper's emphasis on a "unified framework," why not train a single, unified multi-modal model on a combined dataset and evaluate it on both tasks?

2. Regarding the training data, could the authors provide a qualitative comparison between the datasets used by the baselines and the data generated by Euclid-Omni? While the paper emphasizes that the proposed method uses less data, it is possible that this performance gain comes from data specialization rather than just efficiency. Could it be that the baseline datasets are more general or diverse, whereas the Euclid-Omni data is more narrowly tailored to the specific types of problems in the evaluation benchmarks? This would explain why "less data" appears to achieve better results.

3. What is the fundamental difference between Euclidea and existing formal geometry systems like DD+AR and Inter-GPS? The described pipeline of iteratively applying deduction theorems (i.e, rules in some previous works) and updating known quantities via an algebraic solver seems conceptually similar to prior approaches. Is the core contribution of Euclidea a more advanced piece of engineering (e.g., a more comprehensive deductive database and a more precise algebraic system)?

4. The reasoning process in Euclidea involves enumerating inference rules from the deductive database to find applicable theorems. This sounds like it could create a significant efficiency bottleneck, especially as the number of known facts and theorems increases for complex problems.

---

> ### Author Response · Authors · 2025-11-21
>
> Thanks for your review and suggestions! We’d like to address your comments below.
>
> > The completeness and generalization of our symbolic system
>
> The finite set of manually defined theorems is not a fundamental limitation. Theoretically, all plane geometry problems can be derived from a very small axiomatic basis - e.g., Hilbert’s 20 axioms or even Birkhoff’s 4 axioms (built on the real numbers). Thus, completeness does not require an unbounded or ever-growing rule set. Euclidea follows the same philosophy: it formalizes and covers a core set of geometric and algebraic rules, enabling it to generalize to a wide range of problems.
>
> Concretely, Euclidea’s deductive database implements over 170 inference rules (significantly extending the rule sets used in prior systems such as DD+AR and InterGPS), enabling the system to handle both calculation-style and proving problems up to IMO difficulty. Although one can always add more specialized theorems, the current rule set is already sufficiently expressive to cover the vast majority of geometric problems found in high-school curricula, standard examinations, and Olympiad-style competitions. Our near-perfect performance on Geometry3K empirically supports this claim.
>
> For Euclidea’s algebraic system, our equation classification covers all equation types that arise in real geometry problems, including linear, log-linear, and complex forms. Note that the ''complex'' category includes all remaining forms not captured by the first three, making the four-way classification complete.
>
> Although Euclidea is not designed for any specific dataset, its strong performance across both calculation and proving benchmarks demonstrates that it captures a broad and representative portion of real-world geometric reasoning. The remaining hard problems typically require introducing new auxiliary constructions (e.g., adding points or lines), which does not call for additional theorem rules. These steps often demand creative geometric insight, and in practice can be effectively guided by neural models (e.g., LLMs).
>
> > Clarification of the ''neuro-symbolic framework'' in our proposed approaches
>
> The ''neuro-symbolic'' nature of our framework comes from (1) the data generation pipeline and  (2) learning-based models (LLMs/VLMs) used for reasoning and problem solving.
>
> Euclid-Omni generates data by sampling symbolic construction rules, solving the resulting symbolic problems with Euclidea, and then translating the symbolic outputs into natural language while rendering the corresponding diagrams - ensuring that all reasoning comes from Euclidea and LLMs are used for translation. Each generated example can contain five aligned components: a symbolic problem, a symbolic proof, a diagram, a natural-language problem, and a natural-language proof.
>
> Depending on which parts of these five components are used as supervision, neural models can be trained either to perform pure natural-language reasoning or to combine with Euclidea in symbolic reasoning. For example, in Task 1 we train a VLM on *<natural-language problem, diagram → natural-language proof>*, so at inference time the neural model performs the entire reasoning process in natural language *without relying on Euclidea*. In Task 2 we train an LLM on *<symbolic problem → symbolic auxiliary constructions>*. During inference, the LLM performs the crucial symbolic-reasoning step of *predicting auxiliary constructions*, which is essential for solving competition-level geometry problems. Euclidea then uses both the original symbolic problem and the LLM-predicted constructions to generate a formal proof. In both settings, the neural components (LLMs/VLMs) contribute to reasoning, rather than serving as translators for Euclidea.
>
> In summary, the integration of neural models with Euclidea in both data generation and inference is what makes our approach genuinely neuro-symbolic. With the generated data, there can also be additional configurations for inference, such as autoformalization (training VLMs on *<natural-language problem, diagram → symbolic problem>*), informal theorem proving (training VLMs on *<natural-language problem, diagram → natural-language proof>*), and diagram understanding (training VLMs on *<diagram → natural-language problem>*). These configurations further illustrate the breadth of neuro-symbolic capabilities supported by our framework and demonstrate its potential as a foundation for a wide range of geometric reasoning tasks.

---

> > ### Author Response · Authors · 2025-11-21
> >
> > > Experiment design choices
> >
> > Our choices of VLMs and LLMs for calculation- and proving-style problems follow the input formats used in existing benchmarks. For calculation benchmarks, the diagram is always part of the problem and often contains essential geometric information that is not fully stated in text. Therefore, models must process both the diagram and the problem statement, and we train VLMs accordingly. In contrast, Olympiad-level theorem-proving benchmarks do not include diagrams in either their natural-language or symbolic form, so we use LLMs that operate solely on textual inputs.
> >
> > Regarding dataset sizes, we limited Task 1 to 10k synthetic examples due to API cost constraints. Task 1 requires generating natural-language proofs, which must be translated from symbolic proofs using an LLM. We rely on Gemini-2.5 Flash for this translation, and producing 10k high-quality samples already incurs several thousand dollars in API cost. We expect that the performance of our trained VLMs for Task 1 would improve further with larger datasets.
> >
> > For Task 2, no LLM-based translation is required because only symbolic data are needed. This allows us to generate a larger 100k dataset. Moreover, Olympiad-level proving problems are considerably more challenging than calculation tasks, so additional data helps ensure that the models can learn the required symbolic patterns effectively.
> >
> > While we agree that training a unified multimodal model capable of handling both tasks simultaneously is feasible using the Euclid-Omni pipeline, doing so would require substantially larger synthetic datasets and additional LLM and compute resources. Euclid-Omni provides a unified pipeline for generating rich, aligned symbolic and natural-language data, and users can flexibly choose which components to use for different geometric reasoning tasks. Building such a unified multimodal model is an exciting direction, and we view it as promising future work.
> >
> > > Evaluation of the synthetic datasets
> >
> > To evaluate the quality of synthetic data for calculation problems, we first compared the types of target goals produced by different approaches. Euclid-Omni can generate problems involving angles, lengths, areas, and variables bound to these quantities, which reflects the structure commonly seen in existing benchmarks. In contrast, prior synthetic-data pipelines (GeoGen, TR-COT, NeSyGeo) do not generate problems involving *areas or variable-binding goals*, resulting in relatively narrower coverage.
> >
> > To further assess diversity, we applied t-SNE to the question text and compared the distributions across synthetic methods using 5k randomly sampled examples from each dataset for a fair evaluation. The results (see link: https://anonymous.4open.science/r/euclid-rebuttal-511A/image.webp) show that our synthetic data is more diverse and uniformly distributed in the text-embedding space, with less overlap. Other datasets display more clustered regions, indicating a higher proportion of similar or repetitive samples.
> >
> > In addition, because the natural-language solutions in existing datasets are all generated by LLMs, we also use *solution token length* as a proxy for problem difficulty. The table below reports the average and median response lengths for each available dataset. Our synthetic dataset yields the longest responses on both metrics, suggesting that its problems are more challenging and involve more complex reasoning.
> >
> > | | Geo170K | GeoGen | TR-CoT | Ours |
> > |----------|---------|--------|--------|------|
> > | Average  | 109.3   | 210.3  | 86.9   | 247.5 |
> > | Median   | 103.0   | 177.0  | 78.0   | 230.0 |
> >
> > Finally, we emphasize that the Euclid-Omni pipeline is highly configurable. Users can adjust difficulty levels, problem structures, and sampling settings to meet specific training or evaluation requirements.

---

> > > ### Author Response · Authors · 2025-11-21
> > >
> > > > Comparison of Euclidea with other formal geometry systems
> > >
> > > Euclidea is a symbolic engine that we implemented from scratch. The key difference between Euclidea and prior systems lies in their *formalizations*. Approaches such as DD+AR rely on a full-angle formalization, which is suitable only for proving problems. This representation does not distinguish an angle from its supplement and therefore cannot correctly support many angle-related or area-related inferences required in calculation-style problems. In contrast, InterGPS is designed specifically for calculation problems, but its formalizations, geometric inference rules, and algebraic capabilities are limited and cannot handle Olympiad-level reasoning.
> > >
> > > Euclidea uses a novel human-like formalization that enables it to support both calculation and proving problems within a unified framework. Its reasoning engine is engineered to scale to Olympiad-level difficulty and can handle a wider range of problems than existing systems. Euclidea also includes an enriched set of more than 170 geometric inference rules, all defined based on our formalization (compared with 43 in DD+AR and 17 in InterGPS), along with a novel and robust algebraic system for handling complex equations.
> > >
> > > To provide a clearer comparison among formal geometry solvers, we include a table below summarizing their capabilities and differences.
> > >
> > > |    | Calculation | Proving | Human-Like Notations | Able to Solve Competition-Level Problems | Able to Handle Inaccurate Diagrams|
> > > |-----------|:-----------:|:-------:|:--------------------:|:----------------------------------------:|:----------------------------------------:|
> > > | InterGPS  | ✓           | ✗       | ✓                    | ✗                                          | ✓                                        |
> > > | DD+AR     | ✗           | ✓       | ✗                    | ✓                                          | ✗                                        |
> > > | Euclidea  | ✓           | ✓       | ✓                    | ✓                                          | ✓                                        |
> > >
> > > > Efficiency of Euclidea
> > >
> > > Euclidea uses a highly efficient SQL-based deductive database, which is implemented in C and optimized for fast query operations. Our profiling shows that the deductive database accounts for less than 30% of total runtime and is not a performance bottleneck. On the InterGPS benchmark, Euclidea achieves an average/median solving time of 18.45s / 5.70s for the problems it solves. On the Olympiad-level proving problems in JGEX-AG-231,the average/median solving time is 22.57s / 13.53s. These results demonstrate that Euclidea remains efficient across both calculation and proving problems.

---

> > > > ### Comment · Reviewer_qtUi · 2025-11-26
> > > >
> > > > I have read the authors' rebuttal. All my concerns are addressed. Therefore, I will increase the score accordingly.

---

> > > > > ### Author Response · Authors · 2025-11-27
> > > > >
> > > > > Thank you for your follow-up. We appreciate your acknowledgment that our rebuttal addressed your earlier concerns. We also noticed that the updated rating remains below the acceptance threshold, so if there are any remaining points we could clarify or any further improvements that might help you feel more confident and positive about our work, we would be glad to do so. We will revise our paper with additional explanations, experimental results, and related-work discussions. Please feel free to let us know if there is anything specific you would like us to elaborate on.

---

### Official Review · Reviewer_pzHX · 2025-10-30

**Soundness:** 3
**Presentation:** 3
**Contribution:** 3
**Rating:** 6
**Confidence:** 2

**Summary:**

This manuscript introduces a unified neuro-symbolic framework named Euclid-Omni, designed to solve Euclidean geometry problems. The framework combines LLMs and VLMs with a novel, versatile symbolic geometry solver called Euclidea. The Euclidea solver, which is the core of this framework, fuses logical deduction (a deductive database) with algebraic computation (an algebraic engine) to automatically generate human-readable reasoning steps, enabling it to handle both proving-style and calculation-style geometry problems. Furthermore, the manuscript constructs a comprehensive data generation pipeline. This pipeline can automatically synthesize symbolic problems, render corresponding diagrams, generate solution steps using the Euclidea solver, and finally translate this symbolic content into natural language . The resulting synthetic datasets can be used to train LLMs and VLMs.

**Strengths:**

The manuscript is well-structured and content-rich, constructing a complex neuro-symbolic framework that represents a significant workload. The method demonstrates high efficiency in solving Olympiad-level problems. Compared to the significant overhead of AlphaGeometry, Euclid-Omni's hybrid system achieves competitive performance using only a small number of training samples, reducing the training data by orders of magnitude.

**Weaknesses:**

1. In Section 3.1, the problem formalization relies on both "metric relations" and "diagrammatic relations". The paper states that diagrammatic relations (e.g., SameSide(a, b, c, d)) are "implicit but can be extracted from the diagram" . This extraction process is a critical, non-trivial step, yet its implementation is not detailed. How is this extraction performed?
2. The algebraic system in Euclidea (Section 3.1) categorizes equations into four types. While the first three (linear and log-linear) are solved systematically via Gaussian elimination , the handling of the fourth "complex" category (e.g., trigonometric or higher-order polynomial relations) is vague. The paper states simplification and substitution work "in many cases". This raises a key question: What happens when a complex equation cannot be simplified by this method? Does the system abandon that reasoning branch, or are there fallback mechanisms?
3. The claim of generating "human-readable" reasoning steps is made throughout the paper. However, the examples provided in Appendix C.1 consist of highly symbolic, step-by-step logical and algebraic derivations . While these proofs are more compact than those from PyEuclid (e.g., 13 steps vs. 32) , they are not "human-readable" in the sense of a prose proof and remain difficult for a non-expert to parse. How do the authors define and evaluate "human-readability"?
4. In Section 4.2, the VLM is trained on a mixed dataset of 10K synthetic samples from Euclid-Omni and 10K samples from Geo170K . This introduces a significant confounding variable. The strong performance reported could be largely attributable to the existing Geo170K dataset, which the paper notes was added to "better match the out-of-distribution diagrams present in existing benchmarks". To properly assess the contribution of the proposed data generation pipeline, a crucial ablation study is missing. What is the performance of the VLM when trained only on the 10K (or 20K) samples generated by Euclid-Omni?
5. The entire framework's success in training models relies on the quality of its synthetic data (Section 3.2). However, the paper does not provide a clear validation of this synthetic data's quality. While the generation pipeline is described, what measures or standards were used to assess the quality, diversity, and difficulty distribution of the resulting problems?

**Questions:**

Details see the Weaknesses.

---

> ### Author Response · Authors · 2025-11-21
>
> Thanks for your review and suggestions! We’d like to address your comments below.
>
> > How to extract the diagrammatic relations?
>
> If the diagram provides *exact numerical coordinates* for each point (for example, diagrams from GeoGebra or from our symbolic generator), we extract diagrammatic relations directly by computing from these coordinates. For instance, given points `A(0,0)`, `B(1,0)`, `C(1,1)`, and `D(0,1)`, it is straightforward to determine relations such as `SameSide(A, B, C, D)`.
>
> If the diagram does *not contain explicit coordinates* (as in the Geometry3K dataset), we can apply an OCR module to estimate the approximate point positions and then extract the diagrammatic relations based on these estimates.
>
> Importantly, our reasoning engine does *not* rely on perfectly accurate diagrams nor on extracting every possible diagrammatic relation. Missing several diagrammatic relations is also acceptable, and Euclidea can still solve a wide range of problems (including those in Geometry3K) using the available information.
>
>
> > How to solve the complex equations?
>
> After substituting all relations derived from the previous three systems into each complex equation, many equations can be reduced to expressions involving one or two variables. In these cases, we use SymPy to solve the simplified equation and obtain either the value of the single variable or new relations between the two variables.
>
> If an equation cannot be solved at its current stage, we keep it in the algebraic system rather than discarding it. Additional equations introduced later may provide new information that enables a successful substitution or simplification. It is also worth noting that the algebraic system is typically overdetermined - the number of equations exceeds the number of variables. As a result, it is not necessary to solve every equation explicitly in order to derive the target goal.
>
> > Define and evaluate ''human-readability''.
>
> Thanks for raising this question! By *human-readability*, we refer to whether the produced symbolic proofs use human-like notations and inference steps. Full-angle based solvers (such as DD+AR and Newclid) represent angles as pairs of lines and do not distinguish an angle from its supplement in their deductive process or final solutions (like angle (AB-CD)=pi/3 mod pi). This representation *departs from standard geometric notation and is rarely used by humans*, making these systems more tailored to pure proving problems rather than general geometric reasoning.
>
> In contrast, Euclidea’s formalization and generated proofs follow conventional human-style annotations for geometric objects, including lines, angles, and areas, and all inference rules are designed around this representation. While we acknowledge that Euclidea’s raw symbolic proofs are not polished prose in the way a human might write them, they already adhere to human-understandable notation. Moreover, readability can be further improved through template-based refinement and hybrid methods that combine templates with LLM rewriting (as used in Euclid-Omni), resulting in proofs that are more natural and prose-like in practice.
>
> > Performance of VLMs on the purely synthetic dataset
>
> It is worth noting that many geometry benchmarks (e.g., GeoQA) contain word-problem-style diagrams, where scenes include non-geometric objects such as buildings, trees, or tables. Although these tasks ultimately reduce to geometric abstractions, VLMs may first learn to interpret and abstract these visual elements. To equip models with this capability, we include 10k samples from Geo170K—a widely used dataset whose diagrams contain such objects.
>
> To further evaluate VLM performance purely on Euclid-Omni synthetic data, we also fine-tune Qwen2.5-VL-7B-Instruct exclusively on our 10k synthetic dataset. The results (table below) show that training on pure Euclid-Omni data already yields strong performance across multiple benchmarks, outperforming the base model and even surpassing 20K Geo170K training on two datasets. This demonstrates that the model can effectively learn from the clean, symbolically verified, and diverse synthetic examples generated by Euclid-Omni.
>
> |               | GeoQA | Geometry3K | MathVista | MathVerse |
> |-----------------------------------|:-----:|:-----:|:---------:|:---------:|
> | Base model (no fine-tuning)              | 69.4  | 56.4  |   72.2    |   44.1    |
> | Geo170k 10k + Ours 10k            | 76.6  | 61.0  |   74.7    |   51.0    |
> | Geo170k 20k                       | 75.0  | 57.8  |   70.1    |   50.4    |
> | Ours 10k                          | 74.7  | 63.6  |   72.1    |   50.4    |

---

> > ### Author Response · Authors · 2025-11-21
> >
> > > Evaluation of the synthetic datasets
> >
> > To evaluate the quality of synthetic data for calculation problems, we first compared the types of target goals produced by different approaches. Euclid-Omni can generate problems involving angles, lengths, areas, and variables bound to these quantities, which reflects the structure commonly seen in existing benchmarks. In contrast, prior synthetic-data pipelines (GeoGen, TR-COT, NeSyGeo) do not generate problems involving *areas or variable-binding goals*, resulting in relatively narrower coverage.
> >
> > To further assess diversity, we applied t-SNE to the question text and compared the distributions across synthetic methods using 5k randomly sampled examples from each dataset for a fair evaluation. The results (see link: https://anonymous.4open.science/r/euclid-rebuttal-511A/image.webp) show that our synthetic data is more diverse and uniformly distributed in the text-embedding space, with less overlap. Other datasets display more clustered regions, indicating a higher proportion of similar or repetitive samples.
> >
> > In addition, because the natural-language solutions in existing datasets are all generated by LLMs, we also use *solution token length* as a proxy for problem difficulty. The table below reports the average and median response lengths for each available dataset. Our synthetic dataset yields the longest responses on both metrics, suggesting that its problems are more challenging and involve more complex reasoning.
> >
> > | | Geo170K | GeoGen | TR-CoT | Ours |
> > |----------|---------|--------|--------|------|
> > | Average  | 109.3   | 210.3  | 86.9   | 247.5 |
> > | Median   | 103.0   | 177.0  | 78.0   | 230.0 |
> >
> > Finally, we emphasize that the Euclid-Omni pipeline is highly configurable. Users can adjust difficulty levels, problem structures, and sampling settings to meet specific training or evaluation requirements.

---

### Official Review · Reviewer_Q1PA · 2025-10-31

**Soundness:** 2
**Presentation:** 2
**Contribution:** 4
**Rating:** 4
**Confidence:** 4

**Summary:**

This paper proposes Euclid-Omni, a unified neuro-symbolic framework for Euclidean geometric problem solving. The main contribution of this manuscript lies in two parts.
1. A problem solver for proofs and calculations (Euclidea). The workflow of Euclidea contains three sub-modules. a) The problem formulation module transforms natural language text and visual diagram into symbolic formulations (which is defined as “problem state” in the manuscript; b) The Reasoning Engine leverages the rules in an associated deductive database (SQL database) to calculate or prove the goal; and c) The solution generation module translate the complete solution generated by the reasoning engine into human-readable reasoning traces.
2. A Data synthesis pipeline (Euclid-Omni) with four steps: a) Synthetic problem generation step leverages artificial construction rules to generate symbolic questions; b) The diagram rendering step generate a diagram according to the synthetic problem; the c) natural language translation step transforms the symbolic question into natural language via verified templates and refine them with an LLM into fluent statements, step-by-step solutions, and MCQ variants with plausible distractors; d) the task-specific packaging stem assemble calculation-style problems for VLMs and formal language problems for LLMs.

The manuscript claims that symbolic solvers trained on the synthetic dataset can address Olympiad-level problems.

Note: although the authors mentioned Euclid-Omni in the title, it can also be interpreted as the name of a data synthesis method as mentioned in lines 257-260.

**Strengths:**

- Despite having moved a considerable amount of content to the appendices, the paper is generally easy to follow. The figures can help readers comprehend the proposed pipelines.
- The manuscript provides solid technical contributions. Either Euclidea or Euclid-Omni (the data synthesis pipeline) is sufficient to support an individual publication.
- The proposed Euclidea framework shows to be significant, since it shows to be the first one that can simultaneously solve calculation and the 2 proving tasks.

**Weaknesses:**

- The current organization of the manuscript feels somewhat overcrowded: the Methods section takes up too much space, which compresses the room available to clearly describe the experimental setup and results.
- The current manuscript lacks in-depth analyses of experimental results. Apart from knowing that the proposed method achieves better results, this reviewer is interested in knowing how the proposed method achieves such desirable results. The combined Experimental Results in Sections 4.1–4.3 amount to about half a page, and they are mostly descriptive statements of facts.
- This manuscript lacks adequate ablation studies. It is hard for us to attribute the better experimental results to the proposed Euclidea solver and the quality/comprehensiveness of the synthetic data.
- The results in Table 2 are not comparable to some perspectives, since the authors trained the baselines with the synthetic data, but the results of baseline methods “are taken directly from prior works when available”.

**Questions:**

- This reviewer suggests the authors to separate Euclidea and Euclid-Omni (the data synthesis pipeline) into two separate publications. By doing so, the authors will no longer be necessary to delay some key details in their proposed method to the appendices, which this reviewer believes, will significantly enhance the reading experience of potential readers.

This reviewer respects the decision of the authors and acknowledges the technical contributions they have made.

- This reviewer requests the authors to add the following experiments:
    - Use the same set of synthetic data to re-conduct training for Euclidea and the corresponding baseline methods.
    - Use different datasets to train Euclidea, and compare the results with the one trained on the proposed synthetic dataset.
    - Try removing/replacing the implementation method for the three modules/steps in Euclidea
    - Try removing/replacing the implementation for each of the 4 steps in the proposed data synthesis workflow.

---

> ### Author Response · Authors · 2025-11-21
>
> Thanks for your review and suggestions! We’d like to address your comments below.
>
> >  Improving the organization of the manuscript.
>
> Thank you for recognizing the technical contributions and for your suggestions on organization. Both Euclidea and Euclid-Omni are carefully designed with substantial engineering effort, so we placed many implementation details and examples in the appendices for clarity and readability. In the revision, we will further streamline the presentation to keep the implementation more focused.
> Because Euclidea and Euclid-Omni are tightly integrated - and because the trained neural models can perform reasoning in both symbolic and natural language - we chose to present them together to highlight the ''unified neuro-symbolic'' nature of our framework. Some qualitative comparisons were moved to the appendix due to space constraints. In the revision, with the additional allowed page, we will relocate more experimental results to the main text to improve accessibility.
>
> > Ablation studies on the components of Euclidea.
>
> We conducted ablation experiments to evaluate Euclidea when using only the deductive database or only the algebraic system. We ran these variants on both Geometry3K and JGEX-AG-231, and the results (Table below) show that each component alone can solve only a limited subset of problems. In contrast, combining the deductive database and the algebraic system leads to a substantial improvement, demonstrating that the two components are highly complementary.
>
> |       | Deductive Database | Algebraic System | Euclidea |
> |---------------|:------------:|:---------:|:--------:|
> | Geometry3K    |      1       |    36     |   595    |
> | JGEX-AG-231   |     74       |     2     |   207    |
>
> > Ablation studies on the synthetic pipeline in Euclid-Omni without Euclidea.
>
> Euclidea is essential for generating verified symbolic solutions in the Euclid-Omni data generation pipeline. Without Euclidea, one would have to rely on vanilla LLMs to produce solutions, yet these solutions offer no guarantee of correctness. To highlight the importance of Euclidea in ensuring solution validity, we conducted an ablation study on the data generation pipeline.
>
> We randomly sampled 1k natural-language problems from our synthetic dataset and directly asked Gemini-2.5 Flash to solve them. We then checked whether the final numerical answers matched the ground-truth values. The results show that Gemini correctly solved only *69.6%* of the problems, indicating that a large portion of purely LLM-generated solutions are incorrect.
>
> In contrast, when Euclidea is used to generate symbolic solutions and Gemini is used only for natural-language translation, the resulting answers are *100%* consistent with the ground-truth values. This demonstrates that Euclidea is essential for producing reliable symbolic solutions and for maintaining correctness throughout the Euclid-Omni synthetic pipeline.
>
> > Training VLMs with different synthetic data.
>
> Most methods in Table 2 fine-tune Qwen2.5-VL-7B-Instruct as the base model, which is the same backbone used in our experiments. Thus, the main variation across approaches comes from the synthetic training data rather than architectural differences. Because several prior works do not release their datasets, we report their published results directly.
>
> To perform a controlled ablation and enable a fair comparison, we further fine-tuned Qwen2.5-VL-7B-Instruct on several available datasets under matched settings. Specifically, we trained models on: a 20K subset of Geo170K, a mixture of 10K Geo170K + 10K GeoGen samples, a mixture of 10K Geo170K + 10K TR-CoT samples, a 20K GeoGen subset, and a 20K TR-CoT subset. We additionally trained a model on our 10K synthetic problems for an ablation study.
>
> The results (Table below) show that, under comparable compute, Euclid-Omni data consistently yields competitive or superior performance, especially on the Geometry3K dataset. This demonstrates that high-fidelity, symbolically verified constructions, rather than simply increasing dataset size, are essential for improving VLM performance on geometry reasoning benchmarks.
>
> |               | GeoQA | Geometry3K | MathVista | MathVerse |
> |-----------------------------------|:-----:|:-----:|:---------:|:---------:|
> | Base model (no fine-tuning)              | 69.4  | 56.4  |   72.2    |   44.1    |
> | Geo170k 10k + Ours 10k            | 76.6  | 61.0  |   74.7    |   51.0    |
> | Geo170k 20k                       | 75.0  | 57.8  |   70.1    |   50.4    |
> | Geo170k 10k + GeoGen 10k           | 75.3  | 59.8  |   62.9    |   44.9    |
> | Geo170k 10k + TR-CoT 10k            | 72.8  | 55.2  |   74.0    |   44.5    |
> | GeoGen 20k                        | 73.1  | 45.1  |   63.6    |   40.2    |
> | TR-CoT 20k                        | 75.0  | 57.3  |   72.1    |   47.8    |
> | Ours 10k                          | 74.7  | 63.6  |   72.1    |   50.4    |

---

> > ### Author Response · Authors · 2025-11-21
> >
> > > Analysis of synthetic data.
> >
> > To evaluate the quality of synthetic data for calculation problems, we first compared the types of target goals produced by different approaches. Euclid-Omni can generate problems involving angles, lengths, areas, and variables bound to these quantities, which reflects the structure commonly seen in existing benchmarks. In contrast, prior synthetic-data pipelines (GeoGen, TR-COT, NeSyGeo) do not generate problems involving *areas or variable-binding goals*, resulting in relatively narrower coverage.
> >
> > To further assess diversity, we applied t-SNE to the *question text* and compared the distributions across synthetic methods using 5k randomly sampled examples from each dataset for a fair evaluation. The results (see link: https://anonymous.4open.science/r/euclid-rebuttal-511A/image.webp) show that our synthetic data is more diverse and uniformly distributed in the text-embedding space, with less overlap. Other datasets display more clustered regions, indicating a higher proportion of similar or repetitive samples.
> >
> > In addition, because the natural-language solutions in existing datasets are all generated by LLMs, we also use *solution token length* as a proxy for problem difficulty. The table below reports the average and median response lengths for each available dataset. Our synthetic dataset yields the longest responses on both metrics, suggesting that its problems are more challenging and involve more complex reasoning.
> >
> > | | Geo170K | GeoGen | TR-CoT | Ours |
> > |----------|---------|--------|--------|------|
> > | Average  | 109.3   | 210.3  | 86.9   | 247.5 |
> > | Median   | 103.0   | 177.0  | 78.0   | 230.0 |
> >
> > Finally, we emphasize that the Euclid-Omni pipeline is highly configurable. Users can adjust difficulty levels, problem structures, and sampling settings to meet specific training or evaluation requirements.

---

### Official Review · Reviewer_NnSX · 2025-11-10

**Soundness:** 4
**Presentation:** 4
**Contribution:** 4
**Rating:** 10
**Confidence:** 3

**Summary:**

This work introduces Euclid-Omni, a unified neuro-symbolic framework for solving geometric proving and calculation at IMO-level performance.
As in AlphaGeometry, a famous and powerful geometric prover, Euclid-Omni involves a symbolic deduction engine, a language model to propose auxiliary construction, and a data generator to synthesize pre-training data.
Better than AlphGeometry, Euclid-Omni uses much less training data and supports calculation-based problems.
Besides, Euclid-Omni can also take natural-language and diagram inputs and will release the whole framework.
Extensive experiments show that (1) the improved symbolic deduction engine outperforms existing ones (2) Euclid-Omni achieves comparable or superior performance on problems with natural language and diagram as inputs (3) Euclid-Omni achieves performance comparable to AlphaGeometry on IMO-level problems with orders of magnitude less training data.

**Strengths:**

1. The motivation is very solid and exciting. AlphaGeometry-like geometric provers achieve IMO-level performance and are interpretable and verifiable. This work not only extends them to calculation-based problems but also supports natural language and diagrams as inputs. More importantly, those works do not open-source some key components; hence, it is hard for the research community to follow up. If authors release the whole framework (better with data) as promised, it will be very helpful for the research community.
2. The authors carried out considerable work, notably refining the symbolic deduction engine, synthesizing data, and training both VLMs & LLMs.
3. Related works are comprehensively discussed and clearly written.
4. Experiments are comprehensive.

**Weaknesses:**

As a suggestion, instead of a weakness, incorporating a symbolic engine when taking natural language and diagrams as inputs would be better, in terms of interpretability and verifiability.

In my opinion, this is already an excellent work.

**Questions:**

1. How is the completeness (in terms of formal logic) of the rules in the deduction engine?

---

> ### Author Response · Authors · 2025-11-21
>
> Thanks for your review and suggestions! We’d like to address your comments below.
>
> > Incorporating a symbolic engine when taking natural language and diagrams as inputs would be better, in terms of interpretability and verifiability.
>
> We fully agree - this is an exciting direction for future work! This is closely related to *autoformalization*: translating textual descriptions and visual diagrams into formal relations/goals that Euclidea can reason over. Achieving this reliably would require OCR-style geometric parsing from VLMs (e.g., extracting points and their diagrammatic relations). In our current work, we focus on natural-language reasoning for calculation-style tasks to align with existing baselines. However, we believe that combining autoformalization with symbolic deduction, or integrating natural and formal language reasoning, is a promising direction that could further improve interpretability and verifiability. Importantly, our Euclid-Omni pipeline can also generate synthetic data tailored to this setting.
>
> > How is the completeness (in terms of formal logic) of the rules in the deduction engine?
>
> Thank you for raising this important point. When designing Euclidea’s deduction engine, we prioritized practical efficiency and scalability over full formal completeness. Consequently, the current deduction engine is not logically complete in the formal sense. This is primarily due to two design choices:
>
> Euclidea obtains diagrammatic relations directly from the diagram (via numerical coordinates or OCR) during formalization, rather than attempting to infer all such relations via deduction rules.
>
> Euclidea omits a range of inference rules for degenerate or special cases (e.g., zero angles, zero areas, coincident or overlapping objects), which are included in fully complete systems such as the *conceptual* system E [1]. While these rules are essential for theoretical completeness, they rarely arise in typical geometry problems, so we exclude them in the current implementation.
>
> Although incorporating these additional rules would move the system toward formal completeness, they offer limited practical benefit in our target setting and would substantially expand the search space. Therefore, while extending the rule set to achieve system-E-level completeness is straightforward in principle, we focus in this work on practical geometric reasoning tailored to the non-trivial problems in existing datasets.
>
> [1] Avigad, J., Dean, E., and Mumma, J., 2009. A formal system for Euclid’s Elements. The Review of Symbolic Logic, 2(4), pp.700-768.

---

### Author Response · Authors · 2025-11-26

Dear Reviewers,

As the December 2 deadline for rebuttal responses approaches, we wanted to check in regarding any follow-up questions or clarifications you may need on our submission. We would greatly appreciate the opportunity to address any remaining concerns and provide additional explanations if helpful. If possible, we kindly invite you to share your responses at your earliest convenience.

Thank you again for your time and thoughtful reviews.

Best,

The Authors

---

### Meta-Review · Area_Chair_mPHK · 2025-12-24

**Summary:**

The paper proposes Euclid-Omni, a framework integrating a symbolic geometry solver with VLMs via a synthetic data pipeline. While the engineering effort to build the symbolic solver and the data generation pipeline is acknowledged, the consensus among experienced reviewers leans towards rejection.

**Reviewer NnSX’s score of 10 was heavily discounted** in this decision process due to the reviewer's self-admitted lack of expertise (low confidence), time constraints, and the superficiality of their review, which failed to identify critical methodological issues raised by others.

The remaining valid reviews highlight that the system **relies primarily on manually engineered rules rather than novel learning mechanisms**, making the "neuro-symbolic" framing misleading. **Furthermore, the experimental evaluation lacks solidity.** The reported improvement over AlphaGeometry on IMO-AG-30 is statistically insignificant (22 vs. 21 solved) given the small sample size, and the unexplained omission of key baseline results (e.g., AlphaGeometry 20M on JGEX-AG-231) prevents a comprehensive comparison. These experimental flaws, combined with persistent issues regarding paper organization and the incremental nature of the contribution, limit its suitability for ICLR.

**Reviewer Concerns:**

**Addressed:**

- **Data leakage/confounding factors:** The authors provided new experiments training VLMs solely on their synthetic data, effectively addressing Reviewer pzHX's concern that performance was driven by the inclusion of the Geo170K dataset.
- **Ablation studies:** The rebuttal included necessary ablations demonstrating the contribution of the deductive and algebraic components, satisfying Reviewer Q1PA's technical request.

**Still outstanding :**

- **Misleading framing & limited novelty:** Reviewer qtUi and others maintain that the term "neuro-symbolic" is overstated. The core reasoning is purely symbolic/rule-based, with neural networks acting merely as translators. This represents a system integration effort rather than a fundamental machine learning advancement.
- **Paper structure & presentation:** Reviewer Q1PA noted that the manuscript is overcrowded, with critical methodology and experiments relegated to appendices. The rebuttal's clarifications could not retroactively fix the poor reading experience of the submission.
- **Subjectivity of claims:** The claim of "human-readability" (Reviewer pzHX) remains contentious. The output is technically symbolic traces, which differs significantly from the natural language proofs implied by the paper's pitch.

**Reviewer Scores:**

**Reviewer qtUi (2 -> 4):** Upgraded. The reviewer explicitly stated they would increase their score after the rebuttal clarifications. However, the score remained "below the acceptance threshold," indicating a move to 4 rather than a positive recommendation. The reviewer remains unconvinced about the fundamental "neuro-symbolic" contribution.

**Reviewer Q1PA (4):** Unchanged. While the authors provided the missing data, the reviewer's primary concern regarding the "overcrowded" organization and the displacement of key content to the appendix remains a fundamental presentation flaw that hampers the paper's acceptance.

**Reviewer NnSX (10):** **Discounted / unreliable.** This review exhibits characteristics of a "rush job" by a first-time reviewer. The lack of substantive critique, combined with a "Strong Accept" rating despite low confidence (3) and admitted time pressure, renders this score an outlier that does not reflect the paper's actual scientific merit.

**Reviewer pzHX (6):** Unchanged. The reviewer likely appreciates the technical fix regarding the training data but remains marginally above due to the subjective nature of the "human-readable" claims and the incremental contribution over existing symbolic solvers.

---

### Decision · Program_Chairs · 2026-01-26

Reject